# FORGET MANY, FORGET RIGHT: SCALABLE AND PRECISE CONCEPT UNLEARNING IN DIFFUSION MODELS

**Kaiyuan Deng[1], Gen Li[2], Yang Xiao[3], Bo Hui[3], Xiaolong Ma[1]**

[1]The University of Arizona, [2]Clemson University, [3]The University of Tulsa

`kaiyuan0415@arizona.edu`

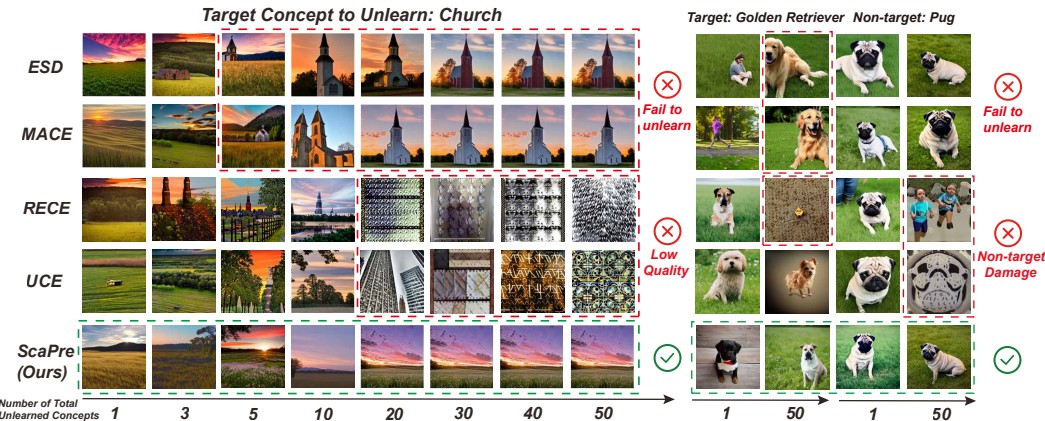

Figure 1: As the total number of unlearned concepts increases, existing methods either fail to effectively forget the target concepts or suffer from severe and noticeable degradation in overall image generation quality. In contrast, our proposed method **ScaPre** consistently maintains both stable unlearning performance and high generative quality (left). Moreover, **ScaPre** does not compromise similar non-target concepts, thereby clearly demonstrating its precise unlearning capability (right).

## ABSTRACT

Text-to-image diffusion models have achieved remarkable progress in generating photorealistic content, but their widespread use raises concerns over copyright and misuse. To mitigate these risks, recent work explores *machine unlearning*, While multi-concept unlearning has shown progress, extending to large-scale scenarios remains difficult, as existing methods face three persistent challenges: *(i)* they often introduce conflicting weight updates, making some targets difficult to unlearn or causing degradation of generative capability; *(ii)* they lack precise mechanisms to keep unlearning strictly confined to target concepts, resulting in collateral damage on similar content; *(iii)* many approaches rely on additional data or auxiliary modules, causing scalability and efficiency bottlenecks as the number of concepts grows. To simultaneously address these challenges, we propose *Scalable-Precise Concept Unlearning (ScaPre)*, a unified and lightweight framework tailored for scalable and precise large-scale unlearning. ScaPre introduces a *conflict-aware stable design*, which integrates the spectral trace regularizer and geometry alignment to stabilize the optimization space, suppress conflicting updates, and preserve the pretrained global structure. Furthermore, the *Informax Decoupler* identifies concept-relevant parameters and adaptively reweight updates, ensuring that unlearning is confined to the target subspace without collateral damage. ScaPre yields an efficient closed-form solution, requiring no additional data or auxiliary sub-models, while maintaining both scalability and precision. Comprehensive experiments across large-scale objects, styles, and explicit content benchmarks demonstrate that ScaPre effectively removes target concepts while maintaining generation quality. It can forget up to $\times\underline{\textbf{5}}$ more concepts than the best baseline within the limits of acceptable generative quality, and outperforms existing multi-concept approaches in precision and efficiency, achieving a new state of the art for large-scale unlearning. Code is available at `https://github.com/kaiyuan02415/scapre`.

# 1 INTRODUCTION

Text-to-image (T2I) generative models (Rombach et al., 2022; Ma et al., 2024; Podell et al., 2023; Saharia et al., 2022; Nichol et al., 2021; Ding et al., 2022; Zhou et al., 2022) have achieved remarkable progress in recent years, enabling users to synthesize high-quality and photorealistic images. Representative models, such as Stable Diffusion (Rombach et al., 2022), DALL·E (Ramesh et al., 2022), and Imagen (Saharia et al., 2022), have been widely adopted across diverse domains, showcasing impressive capabilities in content generation. Despite these advances, the large-scale deployment of T2I models has raised pressing concerns regarding copyright infringement, harmful or biased image synthesis, and the potential misuse of sensitive visual (Kazerouni et al., 2022; Xing et al., 2024; Jang et al., 2022; Wang et al., 2024; Nguyen et al., 2022; Weidinger et al., 2021; Brundage et al., 2018; Tolosana et al., 2020; Solaiman & Dennison, 2021). To address these challenges, recent research has focused on the emerging paradigm of *machine unlearning* (Bourtoule et al., 2021; Guo et al., 2019; Graves et al., 2021; Vatter et al., 2023; Xu et al., 2024; Marino et al., 2025; Zhang et al., 2023; Liu et al., 2024; Tu et al., 2025; Feng et al., 2025), whose objective is to selectively remove the influence of specific data from a trained model while retaining its ability to generate all other content.

Existing works on single-concept unlearning have demonstrated the ability to suppress an individual object, style, or identity in diffusion models (Jia et al., 2023; Kumari et al., 2023; Chavhan et al., 2024; Gandikota et al., 2023; Zhang et al., 2024a; Schramowski et al., 2023; Fan et al., 2023; Heng & Soh, 2023). However, these approaches remain inherently limited to single-concept settings and often suffer from degraded performance when extended to multi-concept unlearning. Beyond single-concept settings, several methods have been proposed for multi-concept unlearning (The number of concepts is usually 10-20). Recent works can be broadly grouped into two directions. One leverages finetuning, including LoRA-based refinements, adapter designs, and mask-driven strategies (Lu et al., 2024; Lyu et al., 2024; Li et al., 2025a). The other relies on closed-form updates, applying direct edits to cross-attention matrices. (Gandikota et al., 2024; Gong et al., 2024).

When facing *large-scale* concept unlearning, existing methods universally encounter several critical challenges: *(i)* they often encounter conflicting weight updates, where updates from different concepts interfere with each other. Such conflicts can make some target concepts difficult to erase or lead to excessive changes in unrelated parameters, causing degradation of generative performance. *(ii)* they lack explicit mechanisms to keep the unlearning strictly focused on the target concepts. As a result, unlearning impacts unintentionally affect background or similar concepts, reducing the precision and reliability of unlearning. *(iii)* most approaches (Lu et al., 2024; Li et al., 2025a; Lyu et al., 2024) rely on extra data or additional sub-models and adapters, which significantly increase computational costs as the number of target concepts grows. Despite recent progress, none of the existing approaches has been able to fully overcome these challenges (as shown in Figure 1 and Figure 6).

To completely overcome these challenges, we propose ***Scalable-Precise Concept Unlearning*** (ScaPre), a unified and lightweight framework for scalable and precise large-scale concept unlearning. First, to address the issue of conflicting weight updates during large-scale unlearning, ScaPre introduces a *conflict-aware stable design* that combines a spectral trace regularizer with geometry alignment, effectively regulating inter-concept interactions and preventing conflicting or excessive updates. Second, to address the challenge of imprecise unlearning, ScaPre introduces an *Informax Decoupler* that measures the information coupling between parameters and target concepts, and adaptively scales their updates, ensuring unlearing is concentrated on the targets while safeguarding background and similar concepts. Unlike prior approaches (Lu et al., 2024; Li et al., 2025a; Lyu et al., 2024) that rely on extra data or auxiliary sub-models, ScaPre employs a single closed-form solution that directly updates weights to simultaneously unlearn an arbitrary number of concepts, entirely training-free. Through this unified and lightweight design, ScaPre achieves scalable, stable, and precise unlearning, and comprehensive experiments demonstrate its strong unlearning capability without sacrificing generation quality. Our contributions are summarized as follows:

- **Scalable Unlearning.** A conflict-aware design with spectral trace regularization and geometry alignment ensures stability when unlearning many concepts.
- **High Precision.** The Informax Decoupler captures information relevance to target concepts and adaptively reweights parameter updates, confining unlearning to the intended subspace.
- **Lightweight Design.** Requires no extra data or sub-models, enabling efficient unlearning with low computational overhead, completing the unlearning of 50 concepts in only ***120 seconds***.

- **Excellent Performance.** ScaPre achieves strong unlearning across benchmarks, effectively removing target concepts while maintaining generation quality. It can unlearn up to $\times 5$ more concepts than the best baseline within the limits of acceptable generative quality.

## 2 RELATED WORK

Machine unlearning was proposed to enable models to selectively remove information, motivated by concerns of data privacy, copyright protection, and the moderation of harmful or biased content.With the rapid deployment of large-scale generative models, these issues have become increasingly critical. Existing approaches can be divided into: *single-concept unlearning* and *multi-concept unlearning*.

### 2.1 SINGLE-CONCEPT UNLEARNING

Early approaches to machine unlearning in diffusion models were originally designed for single-concept forgetting and can be broadly categorized into three types. *Fine-tuning methods*, such as FMN (Zhang et al., 2024a), SA (Heng & Soh, 2023), SalUn (Fan et al., 2023), and AC (Kumari et al., 2023), update model weights—typically through negative guidance or cross-attention restyling—to suppress the target distribution. *Weight-editing methods*, including TIME (Orgad et al., 2023) and SPEED (Li et al., 2025b), directly operate on cross-attention or projection parameters to erase target concepts while retaining unrelated knowledge. *Pruning-based methods* focus on efficiency; ConceptPrune (Chavhan et al., 2024) removes skilled neurons, while MS (Jia et al., 2023) leverages weight pruning to narrow the gap between approximate and exact unlearning. SEMU (Sendera et al., 2025) performs efficient single-concept unlearning by updating only a targeted low-rank subspace.

### 2.2 MULTI-CONCEPT UNLEARNING

Beyond single-concept unlearning, recent studies extend to *multi-concept unlearning*. MACE (Lu et al., 2024) scales to hundreds of concepts through multi-LoRA fine-tuning, while SPM (Lyu et al., 2024) employs composable adapters with latent anchoring to mitigate concept erosion. Sculpting Memory (Li et al., 2025a) leverages dynamic masks and concept-aware optimization for stable multi-concept removal and ESD (Gandikota et al., 2023) fine-tunes diffusion model weights with negative guidance. UCE (Gandikota et al., 2024) offers a unified closed-form framework for multi-concept forgetting, and RECE (Gong et al., 2024) achieves efficient closed-form editing with iterative embedding derivation. These approaches mark progress toward scalable and unified multi-concept editing, but are still insufficient for large-scale forgetting, especially when erasing concrete objects.

## 3 PRELIMINARY

Most existing unlearning methods are training-based, where the model parameters $\theta$ are optimized to forget target concepts while preserving unrelated ones. Let $E$ and $P$ denote the sets of target and non-target concepts, respectively. $f_\theta(x_t, c)$ is the output of the diffusion model at time step $t$ conditioned on prompt $c$, $f_{\theta_{old}}$ is the original model, and $g_{erase}(x_t, c_i^{(E)})$ is a null reference output for the $i$-th target concept. The training objective typically combines an erasure term, a preservation term, and a stability regularization:

$$
\begin{aligned}
\min_\theta \quad & \sum_{i=1}^{|E|} \mathbb{E}_{x,t} \| f_\theta(x_t, c_i^{(E)}) - g_{erase}(x_t, c_i^{(E)}) \|_2^2 \\
& + \lambda_1 \sum_{j=1}^{|P|} \mathbb{E}_{x,t} \| f_\theta(x_t, c_j^{(P)}) - f_{\theta_{old}}(x_t, c_j^{(P)}) \|_2^2 + \lambda_2 \| \theta - \theta_{old} \|_2^2,
\end{aligned}
\tag{1}
$$

Recent works instead formulate unlearning as a closed-form editing problem, which avoids costly gradient-based training and offers higher efficiency. Specifically, given the original key/value projection matrix $W_{old}$ in cross-attention layers, the goal is to construct a new matrix $W$ that redirects target concepts while minimally disturbing others. Let $c_i$ be the embedding of the $i$-th target concept with substitute $c_i^*$, and $c_j$ denote embeddings of preserved concepts. The optimization requires that

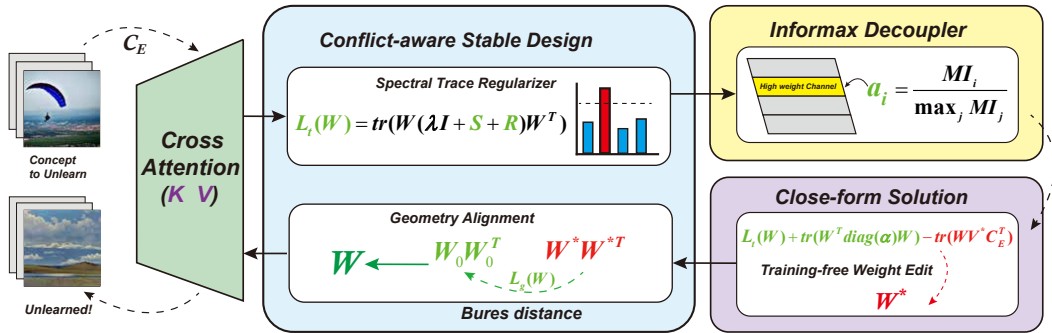

Figure 2: Overview of **ScaPre**. Given target concepts $C_E$, cross-attention representations are first regularized by the spectral trace regularizer $\mathcal{L}_t$, while the informax decoupler $\alpha$ confines updates to concept-relevant subspaces. Within this stabilized space, closed-form optimization yields $W^\star$, realizing the forgetting of $C_E$ (red arrows). Geometry alignment $\mathcal{L}_g$ then applies a proximal refinement toward the pretrained reference $W_0$, preserving global structure (green arrows). This staged pipeline enables scalable, precise unlearning while maintaining non-target generation quality.

target embeddings are mapped to substitute directions, while non-target embeddings remain close to their original outputs. A Frobenius regularization further anchors $W$ to $W_{old}$:

$$\min_W \sum_{c_i \in E} \|W c_i - W_{old} c_i^*\|_2^2 + \lambda_1 \sum_{c_j \in P} \|W c_j - W_{old} c_j\|_2^2 + \lambda_2 \|W - W_{old}\|_F^2. \tag{2}$$

This quadratic objective is convex and admits a closed-form solution obtained by solving the corresponding normal equations (Orgad et al., 2023). Such formulation significantly improves efficiency and reproducibility, since the update can be obtained in a single step without iterative fine-tuning. Owing to these advantages, our framework also builds upon the closed-form paradigm.

## 4 METHOD

In this section, we present ***Scalable-Precise Concept Unlearning (ScaPre)***, a unified framework that achieves both scalability and precision for large-scale unlearning in text-to-image diffusion models (Figure 2). ScaPre addresses two key challenges: instability under large-scale unlearning and insufficient precision in separating target from non-target concepts. Our approach consists of two complementary components. First, we introduce a *conflict-aware stable design* (Sec. 4.1), which leverages a *spectral trace regularizer* and *geometry alignment* to construct a stable optimization space as the number of concepts grows. Second, we develop an *informax decoupler* (Sec. 4.2), which selectively isolates concept-relevant parameters and restricts updates to the corresponding subspace, preventing collateral damage to unrelated knowledge. Finally, these components are combined into a unified optimization objective (Sec. 4.3), yielding a principled solution for scalable and precise unlearning while preserving the generation quality of non-target content.

### 4.1 CONFLICT-AWARE STABLE DESIGN

We propose a conflict-aware stable design with two key components: a ***spectral trace regularizer*** and a ***geometry alignment***. This stage establishes a stable optimization space and maintains overall model stability, preventing instability or degradation during parameter updates.

***Spectral Trace Regularizer.*** When unlearning many concepts simultaneously, conflicting updates can arise and distort the optimization landscape. To mitigate these conflicts and dynamically shape the optimization space, we introduce a *Spectral Trace Regularizer*:

$$\mathcal{L}_t(W) = \text{tr}\big(W(\lambda I + \mathbf{S} + \mathbf{R})W^\top\big), \tag{3}$$

where $W \in \mathbb{R}^{d_{out} \times d_{in}}$ is the key or value projection matrix of the cross-attention module and $\text{tr}(\cdot)$ denotes the trace. Here, $\underline{\lambda I}$ is a standard regularization term commonly used in closed-form solutions (Gandikota et al., 2024; Gong et al., 2024) to ensure numerical stability.

$\mathbf{S}$ captures the dynamic structure of the current concept set by aggregating the second-order statistics of the contextual features of all target concepts:

$$\mathbf{S} = \sum_{k=1}^{m} \sum_{t=1}^{T_k} c_{k,t} c_{k,t}^{\top}, \tag{4}$$

where $m$ is the number of target concepts, $c_{k,t} \in \mathbb{R}^{d_{\text{in}}}$ is the contextual feature vector of the $t$-th token for concept $k$, and $T_k$ is the number of tokens associated with that concept. The span of these features $\{c_{k,t}\}$ identifies the directions most prone to conflicts and noise during large-scale unlearning. By penalizing directions corresponding to large eigenvalues of $\mathbf{S}$, the model adaptively suppresses unstable updates in problematic subspaces.

$\mathbf{R}$ is designed to regulate interactions *within* the target concept subspace. We first construct a matrix of concept embeddings $C_{\text{E}} = [c_1, c_2, \ldots, c_m] \in \mathbb{R}^{d_{in} \times m}$ and perform singular value decomposition (SVD) as $C_{\text{E}} = U \operatorname{diag}(\sigma) V^{\top}$, where each singular value $\sigma_i$ represents the energy along an orthogonal direction. Large $\sigma_i$ indicate directions where multiple concepts strongly overlap and interact. To handle this, we apply a smooth gating function that softly decays the large singular values while leaving smaller ones nearly intact, and reconstruct the matrix as $\mathbf{R} = U \operatorname{diag}(\tilde{\sigma}) U^{\top}$, where $\tilde{\sigma}_i = (1 - \operatorname{sigmoid}(\sigma_i)) \sigma_i$. This adaptive procedure suppresses high-conflict directions caused by overlapping concepts while preserving low-conflict directions associated with independent concepts.

***Geometry Alignment.*** To prevent unlearning from degrading unrelated content, we introduce a geometry alignment term to stabilize the optimization globally. Instead of penalizing raw weight differences with an $\ell_2$ norm like current methods, we treat each row of the cross-attention matrix $W \in \mathbb{R}^{d_{\text{out}} \times d_{\text{in}}}$ as a covariance factor, where $d_{\text{in}}$ and $d_{\text{out}}$ are the input and output dimensions. The pretrained reference is denoted as $W_0$ with the same shape. This defines covariance matrices $WW^{\top}$ and $W_0 W_0^{\top}$, which are aligned using the Bures distance (Bures, 1969; Takatsu, 2008):

$$\mathcal{L}_{\mathbf{g}}(W) = \operatorname{tr}\left(WW^{\top}\right) + \operatorname{tr}\left(W_0 W_0^{\top}\right) - 2\operatorname{tr}\left[\left((WW^{\top})^{1/2} W_0 W_0^{\top} (WW^{\top})^{1/2}\right)^{1/2}\right]. \tag{5}$$

Compared to $\ell_2$ norm, the Bures distance alignment matches the covariance structures of $W$ and $W_0$, rather than only penalizing element-wise weight differences. By preserving these higher-order feature correlations, it maintains the pretrained global structure, keeping unrelated features stable. Combined with the trace regularizer, geometry alignment provides a complementary global safeguard against destructive degradation during large-scale unlearning.

## 4.2 INFORMAX DECOUPLER

The scalable design in Sec. 4.1 establishes a safe and stable optimization space. Building on this foundation, we introduce a decoupler that identifies parameters most relevant to the target concepts and confines updates strictly within the corresponding subspace.

When unlearning multiple concepts, different weights contribute unevenly: some are strongly tied to target concepts, while others mainly support unrelated background. Treating all weights equally may thus cause either incomplete unlearning or unnecessary collateral damage. To address this, we introduce the *Informax Decoupler*, which leverages mutual information (MI) to quantify how strongly each weight is coupled with the target concepts. For each channel $i$, we define a discretized activation state $z = \mathbb{1}\{a_i(s) > \tau_i\} \in \{0,1\}$, where $a_i(s) = W_{i:}s$ is the activation of channel $i$ on input feature $s$, and $\tau_i$ is an adaptive threshold. We further denote by $y \in \{0,1\}$ the input label, with $y = 1$ for target-concept inputs and $y = 0$ for neutral inputs. From the resulting activation–label pairs $\{(z, y)\}$, we estimate the empirical joint distribution as $p_i(z, y) = n_{zy}/K$, where $n_{zy}$ is the number of pairs with activation state $z$ and label $y$, and $K$ is the total sample size. The marginal distributions are given by $p_i(z) = \sum_y p_i(z, y)$ and $p_i(y) = \sum_z p_i(z, y)$. The MI of channel $i$ is then defined as:

$$\text{MI}_i = \sum_{z \in \{0,1\}} \sum_{y \in \{0,1\}} p_i(z, y) \log \frac{p_i(z, y)}{p_i(z) \, p_i(y)}. \tag{6}$$

For each channel $W_{i:}$, MI quantifies how informative its activations are about the input type, with larger $\text{MI}_i$ indicating that the channel is more predictive of whether the input contains target concepts. For multiple target concepts $\{c_k\}_{k=1}^{m}$, we compute a per-concept score $\text{MI}_i^{(k)}$ (using positives from

$c_k$) and aggregate by the maximum over concepts: $\mathrm{MI}_i = \max_k \mathrm{MI}_i^{(k)}$. Finally, we obtain the ***Informax Decoupler***:

$$\boldsymbol{\alpha} = [\alpha_1, \ldots, \alpha_{d_{\mathrm{out}}}], \quad \alpha_i = \frac{\mathrm{MI}_i}{\max_j \mathrm{MI}_j} \in [0, 1]. \tag{7}$$

It maximizes information about the distinction between target and non-target inputs, and thus decoupling concept-relevant parameters from unrelated ones.

### 4.3 SCALABLE-PRECISE CONCEPT UNLEARNING

Having established a stable optimization space and a mechanism to selectively decouple concept-relevant parameters, we now bring these components together into a unified framework. Let $A = \lambda I + \mathbf{S} + \mathbf{R}$ and $B = \mathrm{diag}(\boldsymbol{\alpha})$. The overall objective of ScaPre is formulated as:

$$\min_W \; \mathrm{tr}\left(WAW^\top\right) \; + \; \beta\,\mathcal{L}_{\mathbf{g}}(W) \; + \; \mathrm{tr}\left(W^\top BW\right) \; - \; \mathrm{tr}\left(WV^*C_E^\top\right) \tag{8}$$

Here, $C_E$ denotes the stacked embedding matrix of concepts to erase, and $V^*$ specifies their replacement targets (often set to zero for complete forgetting). The coefficient $\beta$ modulates the geometry alignment, balancing global stability with flexibility for targeted unlearning.

To optimize equation 8, we first ignore the geometry alignment term ($\mathcal{L}_{\mathbf{g}}(W)$) and solve the quadratic part analytically. Taking the derivative with respect to $W$ and setting it to zero yields the Sylvester equation:

$$BW \; + \; WA \; = \; V^*C_E^\top \tag{9}$$

This equation has a unique solution and can be solved efficiently with standard Sylvester solvers. In vectorized form, the solution can be written as:

$$\mathrm{vec}(W^\star) = \left(I_{d_{\mathrm{in}}} \otimes B \; + \; A^\top \otimes I_{d_{\mathrm{out}}}\right)^{-1} \mathrm{vec}(V^*C_E^\top), \tag{10}$$

where $\mathrm{vec}(\cdot)$ denotes the column-wise vectorization operator that stacks all columns of a matrix into a single vector; once $\mathrm{vec}(W^\star)$ is obtained, $W^\star$ can be recovered by reshaping it back to its original $d_{\mathrm{out}} \times d_{\mathrm{in}}$ form. A complete derivation is provided in Appendix B.1.

The geometry alignment term ($\mathcal{L}_{\mathbf{g}}(W)$), however, involves matrix square roots nested inside co-variance operators, which makes the overall objective no longer purely quadratic and therefore incompatible with direct closed-form optimization. As a result, it must be handled separately. We treat $W^\star$ as an intermediate solution and incorporate geometry alignment via a proximal refinement. The refinement proceeds in two conceptual steps: (1) In the covariance space, we replace $\Sigma^\star = W^\star W^{\star\top}$ with a new covariance $\Sigma^+$ obtained by moving partway along the Bures geodesic toward the pretrained reference $\Sigma_0 = W_0 W_0^\top$. Intuitively, this step softly aligns the overall geometry of $W^\star$ with that of the pretrained model. (2) We then map back to the weight space by finding the matrix $\widetilde{W}$ whose covariance equals $\Sigma^+$ and that lies closest to $W^\star$. This is done through an orthogonal Procrustes adjustment, i.e., an orthogonal rotation that preserves the main unlearning direction while enforcing global stability. A full derivation of the exact proximal refinement is provided in Appendix B.2.

## 5 EXPERIMENT

In this section, we describe the experimental setting (Sec. 5.1), evaluate large-scale (Sec. 5.2), precise (Sec. 5.3), and artistic style unlearning (Sec. 5.4), and analyze efficiency (Sec. 5.5). Additional results including adversarial robustness and explicit content are in Appendix C.3 and C.2, a detailed analysis of generative quality in Appendix C.4 and Figure 12, and ablation studies in Appendix C.5-C.7.

### 5.1 EXPERIMENTAL SETTINGS

All experiments were conducted using Stable Diffusion v1.4 & v1.5 (Rombach et al., 2022). Large-scale unlearning is evaluated on the Imagenette benchmark (Howard et al., 2019), a ten-class subset of ImageNet, and on our custom dataset ***ImageNet-Diversi50***, which includes 50 diverse object categories sampled from ImageNet to evaluate scalability and diversity in unlearning. For precise

**Table 1:** Overall comparison of different unlearning methods on Imagenette, reporting average unlearning accuracy ($\downarrow$), CLIP score ($\text{CLIP}_{coco}$) ($\uparrow$), and the unified metric **UQ** ($\uparrow$).

| Metric | Method | | | | | | | | |
|---|---|---|---|---|---|---|---|---|---|
| | SD v1.5 | FMN | SPM | ESD | MACE | UCE | RECE | SP | *ScaPre* (Ours) |
| **Avg Acc** ($\downarrow$) | 89.9 | 71.9 | 47.4 | 38.7 | 78.5 | 8.5 | 4.9 | 9.6 | **0.8** |
| **CLIP**$_{coco}$ ($\uparrow$) | 31.43 | 30.62 | 30.81 | 30.14 | **31.02** | 29.45 | 29.27 | 29.25 | 30.43 |
| **UQ** ($\uparrow$) | — | 37.35 | 49.89 | 47.84 | 35.07 | 37.23 | 32.60 | 31.78 | **64.09** |

**Table 2:** We report evaluation across multiple metrics including $\text{CLIP}_{art}$, $\text{CLIP}_{coco}$, $\text{CLIP}_{x}$ (difference), and FID. ScaPre consistently outperforms baselines.

| Method | CLIP$_{art}$ ($\downarrow$) | CLIP$_{coco}$ ($\uparrow$) | CLIP$_{x}$ ($\uparrow$) | FID ($\downarrow$) |
|---|---|---|---|---|
| SD v1.5 | 31.25 | 31.43 | 0.18 | 13.60 |
| FMN | 30.67 | **31.20** | 0.53 | 14.72 |
| SPM | 29.40 | 29.73 | 0.33 | 22.75 |
| ESD | 27.24 | 27.94 | 0.70 | 17.22 |
| MACE | 27.34 | 30.06 | 2.72 | **13.89** |
| UCE | 26.95 | 28.21 | 1.26 | 43.72 |
| RECE | 25.94 | 26.86 | 0.92 | 49.32 |
| SP | 29.35 | 28.75 | -0.60 | 19.76 |
| **ScaPre (Ours)** | **26.51** | 29.95 | **3.44** | 14.37 |

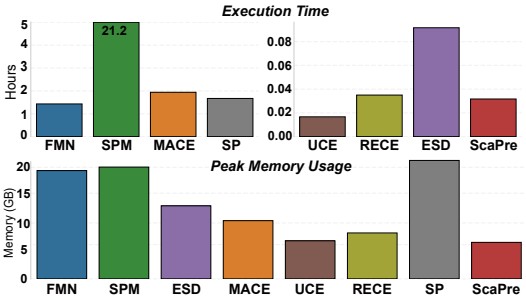

**Figure 3:** Comparison of efficiency across methods, reporting GPU-hours and peak memory usage.

unlearning, we construct ***ImageNet-Confuse5***, consisting of five groups of visually similar concepts from ImageNet that can be reliably generated, enabling us to evaluate fine-grained disentanglement and robustness to concept confusion. Explicit content unlearning is conducted on the I2P dataset (Schramowski et al., 2023), which contains a variety of inappropriate image prompts. For artistic style unlearning, we select 50 artists whose styles can be reliably reproduced by Stable Diffusion. We use the widely recognized MS COCO-30K (Lin et al., 2014) to assess the model's generative performance after unlearning. All experiments are performed on an NVIDIA RTX A6000 GPU. For reproducibility, all results of the compared methods are derived from their official open-source implementations, as listed in Appendix D.

## 5.2 Unlearning at Scale

We begin with unlearning experiments on the widely adopted *Imagenette* (Howard et al., 2019) benchmark. For each target class, we measure *unlearn accuracy* ($\downarrow$) using a pretrained ResNet-50 classifier, where lower values indicate stronger removal of the forgotten concept. To assess the preservation of generation quality, we report the CLIP score ($\uparrow$), measured on the MS COCO dataset, which quantifies the alignment between generated images and text prompts. In addition, we introduce a unified metric **UQ** ($\uparrow$), short for *Unlearn & Quality*, which jointly reflects unlearning effectiveness and generation quality. It harmonically integrates a normalized unlearning score $\tilde{A}$ and a normalized quality score $\tilde{C}$. Here, $\tilde{A} = \sigma((\mu_A - A)/\sigma_A)$ and $\tilde{C} = \sigma((C - \mu_C)/\sigma_C)$, where $\sigma(\cdot)$ is the sigmoid function, $\mu_A$ and $\sigma_A$ are the mean and standard deviation of unlearning accuracy across all methods, and $\mu_C$ and $\sigma_C$ are those of CLIP score. **UQ** is then computed as $UQ = 100 \cdot \frac{2\tilde{A}\tilde{C}}{\tilde{A}+\tilde{C}}$. Larger **UQ** values indicate stronger unlearning with better quality retention. As summarized in Table 1, ScaPre consistently achieves the lowest forgetting accuracy while maintaining competitive CLIP scores. The full set of experimental results, including results for each individual class, is provided in the Appendix C Table 5. We then evaluate ScaPre and a wide range of baselines on the extended benchmark (ImageNet-Diversi50). As shown in Table 3, ScaPre achieves substantially better performance than all competing methods. Figure 5 further illustrates the scalability of our approach when the number of target concepts increases. For UCE and RECE, the curves are truncated because both methods suffer severe generative collapse under large-scale unlearning, making results in that region uninformative. For the other baselines, unlearning performance consistently deteriorates as more concepts are removed, leading to unstable or incomplete unlearning. These findings demonstrate that ScaPre is not only more effective but also substantially more reliable than existing baselines. Complete results are provided in Appendix C, Table 6.

**Table 3:** We demonstrate the overall results on the ImageNet-Diversi50, including Avg Acc, CLIP Score and UQ. ScaPre achieves significantly better performance compared to all baselines.

| Method | Avg Acc (↓) | CLIP$_{coco}$ (↑) | UQ (↑) |
|---|---|---|---|
| SD v1.5 | 87.7 | 31.43 | — |
| FMN | 79.8 | 30.12 | 36.52 |
| SPM | 76.5 | 30.45 | 38.67 |
| ESD | 19.6 | 28.21 | 56.35 |
| MACE | 78.9 | **31.23** | 38.11 |
| UCE | 0.0 | 22.23 | 25.16 |
| RECE | 0.0 | 21.78 | 22.90 |
| SP | 22.5 | 28.83 | 51.28 |
| **ScaPre (Ours)** | **3.9** | 29.41 | **65.30** |

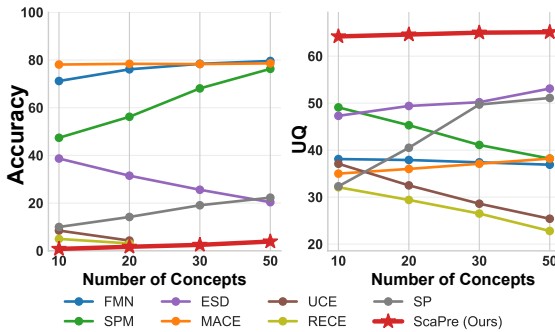

**Figure 4:** We compare how unlearning accuracy (↓) and UQ (↑) change as the number of concepts increases across methods. ScaPre consistently achieves the best performance.

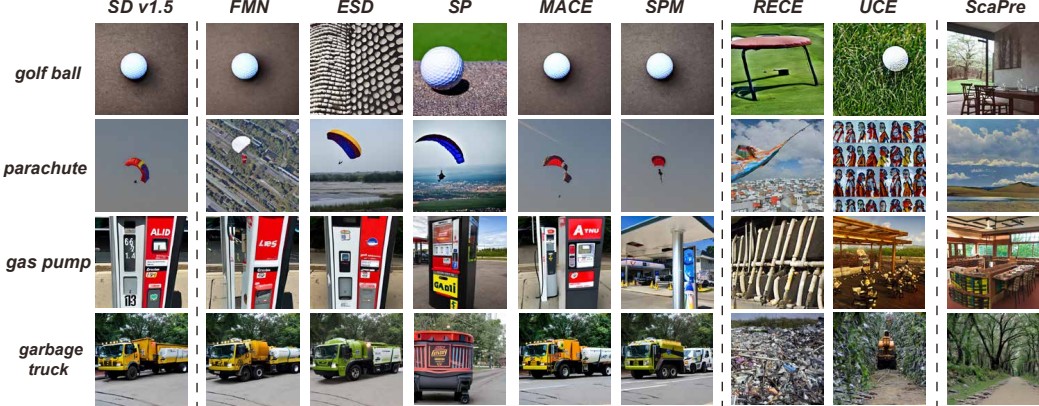

**Figure 5:** Visual comparison of unlearning performance across different methods on the simultaneous unlearning of 10 classes from the Imagenette benchmark.

## 5.3 PRECISE UNLEARNING

To rigorously assess the capacity for precise unlearning in scalable scenarios, we curate ImageNet-Confuse5, a benchmark that contains five groups of visually similar ImageNet concepts. Within each group, two concepts are designated as unlearning targets, while the remaining three are visually similar but non-target concepts that should be preserved. This design makes the task especially challenging, as it requires disentangling highly confusable concepts rather than relying on coarse category differences. Table 4 presents the results across methods. The evaluation considers unlearn accuracy (↓) for residual accuracy on targets, preserve accuracy (↑) for retention on similar but non-target concepts, and ***overall accuracy (↑)*** as their harmonic mean: $\frac{2(100-A)\cdot P}{(100-A)+P}$, with $A$ and $P$ denoting unlearn and preserve Acc, respectively. Compared to prior methods, which either exhibit poor unlearning performance or undesirably suppress related categories, ScaPre achieves the best overall accuracy and UQ, demonstrating superior disentanglement while maintaining high generative fidelity. For complete experimental results, please refer to the Appendix C Table 7.

## 5.4 ARTISTIC-STYLE UNLEARNING

For artistic style unlearning, we select fifty representative artists whose styles can be faithfully reproduced by Stable Diffusion, providing a reliable benchmark for evaluation. We first measureCLIP$_{art}$, which reflects the residual stylistic similarity after unlearning and directly measures forgetting effectiveness. To assess generation quality, we jointly consider CLIP$_{coco}$ and FID on MS-COCO. Finally, we report CLIP$_x$, defined as the difference between CLIP$_{coco}$ and CLIP$_{art}$, as a holistic indicator of the balance between forgetting and quality preservation. As shown in Table 2, ScaPre establishes the

**Table 4:** Overall evaluation summary for **ImageNet-Confuse5**. Unlearn Acc (↓) quantifies residual accuracy on target concepts (described in Table 7), Preserve Acc (↑) measures retention on visually similar but non-target concepts, and **Overall Acc (↑)** reflects the joint ability to both unlearn and preserve. CLIP$_{coco}$ and UQ further evaluate generation quality and unlearning precision.

| Metric | Method | | | | | | | | |
|---|---|---|---|---|---|---|---|---|---|
| | SD v1.5 | FMN | SPM | ESD | MACE | UCE | RECE | SP | *ScaPre* (Ours) |
| Unlearn Acc (↓) | 83.9 | 76.5 | 77.5 | 55.6 | 76.4 | 2.9 | 3.1 | 55.0 | **5.8** |
| Preserve Acc (↑) | 86.6 | 78.9 | 79.7 | 57.7 | 78.6 | 5.6 | 5.5 | 57.1 | **76.3** |
| Overall Acc (↑) | 27.2 | 36.2 | 35.1 | 50.2 | 36.3 | 10.6 | 10.4 | 50.3 | **84.3** |
| CLIP$_{coco}$ (↑) | 31.43 | 30.45 | 30.60 | 29.78 | **30.98** | 28.04 | 27.23 | 29.84 | 30.15 |
| UQ (↑) | 38.27 | 40.29 | 40.28 | 46.69 | 42.13 | 31.88 | 20.41 | 47.47 | **65.49** |

**Figure 6:** Visual comparison of unlearning performance across different methods on the Imagenette benchmark

most favorable trade-off, delivering both more effective unlearning and superior generation quality over prior approaches, while maintaining robustness across diverse styles.

## 5.5 EFFICIENCY OF SCAPRE

A key strength of ScaPre lies in its efficiency, which is particularly critical when scaling unlearning methods to large numbers of concepts and real-world deployments. As illustrated in Figure 3 (We have provided specific data in the Appendix C Table 11), most existing approaches suffer from substantial computational overhead, often demanding both long training time and large memory consumption, yet their overall performance still lags behind. In contrast, ScaPre maintains a lightweight profile with minimal execution time and memory usage, completing the unlearning of 50 concepts within only 120 seconds. Although methods such as UCE and RECE also exhibit low runtime and memory costs, their unlearning effectiveness and generative quality fall far short of ScaPre. Taken together, ScaPre establishes a uniquely favorable trade-off, combining scalability, precision, and efficiency in a way that prior approaches do not achieve.

## 6 CONCLUSION

In this work, we introduced *ScaPre*, the first closed-form framework specifically designed for large-scale concept unlearning in diffusion models. By combining a conflict-aware stable design with an informax decoupler, ScaPre enables stable optimization, precise disentanglement, and highly efficient computation. Our closed-form formulation eliminates the need for fine-tuning and auxiliary modules, achieving scalability to a large number of concepts with minimal overhead. Extensive experiments across objects, styles, and explicit content benchmarks demonstrate both superior unlearning efficacy and consistently high generative fidelity. We believe ScaPre establishes a new and effective paradigm for reliable large-scale unlearning, offering a principled and forward-looking direction for future research in safe and controllable generative modeling.

## 7 ETHICS STATEMENT

This work does not involve human subjects, personally identifiable data, or private information. We therefore believe this work does not raise ethical concerns.

## 8 REPRODUCIBILITY STATEMENT

We have made every effort to ensure that the results presented in this paper are reproducible. A detailed description of the complete algorithmic workflow is provided, and the full derivations are included in the Appendix to facilitate faithful implementation of the entire method. All generative models and baseline methods used in this study are based on publicly available open-source code, ensuring consistency and reproducibility of the evaluations. We will release the full code repository in the near future.

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

## A    LLM USAGE STATEMENT

Large language models (LLMs) were employed solely for polishing the manuscript, including refining clarity and checking/correcting grammatical errors and typos. LLMs did not contribute to the generation of substantive ideas, analyses, or results. All scientific content, interpretations, and conclusions are entirely the authors' own.

## B    DETAILED DERIVATIONS FOR SCAPRE

We provide detailed derivations for the optimization steps in ScaPre. We first present the closed-form solution of the quadratic part of the objective (ignoring geometry alignment), and then describe the proximal refinement procedure used to incorporate Bures alignment. For clarity, all variables, operators, and dimensions are explicitly stated once and used consistently.

### B.1    CLOSED-FORM SOLUTION OF THE QUADRATIC PART

We start from the quadratic component of Eq. equation 8, ignoring the geometry alignment (Bures distance) term:

$$f(W) = \text{tr}\left(WAW^\top\right) + \text{tr}\left(WBW^\top\right) - \text{tr}\left(WM^\top\right), \tag{11}$$

where:

- $W \in \mathbb{R}^{d_{\text{out}} \times d_{\text{in}}}$: weight matrix being updated (cross-attention).
- $A = \lambda I + \mathbf{S} + \mathbf{R} \in \mathbb{R}^{d_{\text{in}} \times d_{\text{in}}}$: input-side stabilizer (symmetric, typically $A \succ 0$).
- $B = \text{diag}(\boldsymbol{\alpha}) \in \mathbb{R}^{d_{\text{out}} \times d_{\text{out}}}$: output-side channel weighting from the Informax Decoupler (symmetric, $B \succeq 0$).
- $M = V^* C_E^\top \in \mathbb{R}^{d_{\text{out}} \times d_{\text{in}}}$: with $C_E \in \mathbb{R}^{d_{\text{in}} \times m}$ the stacked embeddings of the $m$ concepts to erase, and $V^* \in \mathbb{R}^{d_{\text{out}} \times m}$ their replacement targets (often zero).

Using standard matrix calculus for symmetric $A, B$, the first-order optimality condition (after canceling a common factor) simplifies to the Sylvester equation:

$$BW + WA = M. \tag{12}$$

Equation equation 12 has a unique solution if the spectra satisfy $\sigma(B) \cap (-\sigma(A)) = \varnothing$. In our setting, $A \succ 0$ and $B \succeq 0$, hence uniqueness holds.

Using $\text{vec}(BW) = (I_{d_{\text{in}}} \otimes B)\,\text{vec}(W)$ and $\text{vec}(WA) = (A^\top \otimes I_{d_{\text{out}}})\,\text{vec}(W)$, we obtain

$$\left(I_{d_{\text{in}}} \otimes B + A^\top \otimes I_{d_{\text{out}}}\right)\text{vec}(W) = \text{vec}(M).$$

Thus,

$$\text{vec}(W^\star) = \left(I_{d_{\text{in}}} \otimes B + A^\top \otimes I_{d_{\text{out}}}\right)^{-1}\text{vec}(M). \tag{13}$$

Here $\text{vec}(\cdot)$ stacks matrix columns; once $\text{vec}(W^\star)$ is obtained, $W^\star$ is recovered by reshaping to $d_{\text{out}} \times d_{\text{in}}$.

Let $A = Q_A \Lambda_A Q_A^\top$, $B = Q_B \Lambda_B Q_B^\top$ be eigendecompositions. Define $X = Q_B^\top W Q_A$, $\widehat{M} = Q_B^\top M Q_A$. Then

$$\Lambda_B X + X \Lambda_A = \widehat{M},$$

which decouples element-wise:

$$X_{ij} = \frac{\widehat{M}_{ij}}{\lambda_{B,i} + \lambda_{A,j}}.$$

Finally,

$$W^\star = Q_B X Q_A^\top.$$

In our case, $B = \text{diag}(\alpha)$ is already diagonal ($Q_B = I$), so only a single eigendecomposition of $A$ is needed, followed by element-wise division and back-transformation.

## B.2 Proximal Refinement for Geometry Alignment

We now detail the treatment of the geometry alignment (Bures distance) term:

$$\mathcal{L}_g(W) = \text{tr}(WW^\top) + \text{tr}(W_0 W_0^\top) - 2\,\text{tr}\Big[\big((WW^\top)^{1/2} W_0 W_0^\top (WW^\top)^{1/2}\big)^{1/2}\Big], \quad (14)$$

where $W_0 \in \mathbb{R}^{d_{\text{out}} \times d_{\text{in}}}$ is the pretrained reference and $WW^\top$, $W_0 W_0^\top \in \mathbb{R}^{d_{\text{out}} \times d_{\text{out}}}$ are the induced covariances.

Because of nested matrix square roots, jointly optimizing equation 11 and equation 14 in one closed form is infeasible; we therefore apply a proximal refinement after solving the quadratic part.

Compute $W^\star$ via Eq. equation 13 (or the spectral method) and form

$$\Sigma^\star = W^\star W^{\star\top} \in \mathbb{R}^{d_{\text{out}} \times d_{\text{out}}}.$$

Let $\Sigma_0 = W_0 W_0^\top$. Interpolate toward $\Sigma_0$ along the Bures geodesic with weight $\beta \in [0,1]$:

$$\Sigma^+ = \big((1-\beta)\Sigma^{\star 1/2} + \beta(\Sigma^{\star 1/2}\Sigma_0\Sigma^{\star 1/2})^{1/2}\big)^2.$$

This is a geometry-aware interpolation on the PSD cone, not a naive Euclidean average.

Find $\widetilde{W}$ such that $\widetilde{W}\widetilde{W}^\top = \Sigma^+$ and $\widetilde{W}$ is as close as possible to $W^\star$.

Proceed by orthogonal Procrustes:

1. Eigendecompose $\Sigma^+ = U\Lambda U^\top$, take $U\Lambda^{1/2}$.
2. Solve $\min_{Q^\top Q = I} \|W^\star - U\Lambda^{1/2}Q^\top\|_F$.
3. With $K = W^{\star\top}U\Lambda^{1/2} = U_K\Sigma_K V_K^\top$ (SVD), the optimum is $Q^\star = U_K V_K^\top$.
4. Set $\widetilde{W} = U\Lambda^{1/2}Q^{\star\top}$.

This preserves the interpolated covariance while staying closest to $W^\star$ up to an orthogonal rotation.

## C Additional Experimental Results

### C.1 Unlearning at Scale and Precise Unlearning

We provide per-concept unlearning results for all benchmarks reported in the main paper, including the 50-class ImageNet-Diversi50 (Table 6), Imagenette (Table 5), and ImageNet-Confuse5 (Table 7). Across nearly all concepts, ScaPre attains the lowest unlearn accuracy while maintaining competitive CLIP scores and the highest **UQ**, indicating effective removal of target concepts without degrading generation quality. On visually confusable groups, ScaPre yields the best overall accuracy by jointly unlearning targets and preserving similar non-targets, highlighting its precision.

We also provide comprehensive visual results to complement the quantitative analysis. As shown in Figure 8 and Figure 10, under large-scale unlearning condition, existing methods either fail to fully unlearn the target concepts or severely degrade the generative ability of the model. In contrast, ScaPre delivers consistently high-quality and stable generations while achieving precise removal of the target concepts, even in large-scale settings. As shown in Figure 9, the results demonstrate that fine-tuning–based approaches fail to satisfy unlearning requirements when scaled to a large number of concepts. In contrast, UCE and RECE severely disrupt similar but non-target concepts, thereby compromising precision. Our proposed ScaPre, however, achieves precise unlearning of the target concepts while preserving the integrity of closely related non-target concepts, highlighting its effectiveness and reliability in large-scale unlearning scenarios. For artistic styles unlearning, as shown in Figure 7, all existing baselines fail to achieve effective unlearning, whereas ScaPre successfully removes the target styles while preserving high generation quality. We also compare ScaPre with representative baselines that exhibit relatively strong multi-concept unlearning capabilities. As the number of target concepts increases (as shown in Figure 12), these baselines suffer a drastic decline in generative quality, quickly falling below an acceptable range. In contrast, our method maintains consistently high generative fidelity even when unlearning up to 50 concepts. This demonstrates that, under comparable quality requirements, ScaPre can successfully forget more than five times the number of concepts compared to prior baselines.

**Table 5:** Comparison of different unlearning methods on Imagenette benchmark, We report unlearning accuracy (↓), CLIP score (↑) and *UQ* (↑).

| Imagenette classes | Method | | | | | | | | |
|---|---|---|---|---|---|---|---|---|---|
| | SD v1.5 | FMN | SPM | ESD | MACE | UCE | RECE | SP | *ScaPre* (Ours) |
| parachute | 92.8 | 90.6 | 74.5 | 6.5 | 88.9 | 8.3 | 3.0 | 0.2 | **0.0** |
| golf ball | 97.6 | 92.5 | 93.2 | 13.4 | 94.9 | 7.7 | 4.2 | 12.2 | **0.0** |
| garbage truck | 88.7 | 79.2 | 54.1 | 31.4 | 82.5 | 28.7 | 15.3 | 43.2 | **4.0** |
| cassette player | 67.1 | 15.2 | 3.7 | 4.9 | 15.1 | 2.2 | 1.1 | 6.1 | **0.0** |
| church | 86.2 | 77.3 | 66.9 | 61.6 | 69.0 | 19.6 | 13.4 | 21.3 | **4.0** |
| tench | 97.9 | 79.0 | 43.7 | 64.7 | 84.5 | 5.1 | 1.0 | 2.2 | **0.0** |
| English springer | 97.6 | 85.8 | 67.5 | 79.5 | 92.7 | 0.9 | 2.0 | 1.2 | **0.0** |
| French horn | 97.9 | 87.1 | 17.1 | 57.5 | 96.1 | 3.7 | 2.1 | 0.3 | **0.0** |
| chain saw | 77.9 | 43.4 | 32.5 | 12.0 | 73.2 | 4.8 | 3.1 | 4.1 | **0.0** |
| gas pump | 94.2 | 69.4 | 20.7 | 55.2 | 83.5 | 5.3 | 4.2 | 5.2 | **0.0** |
| Avg Acc (↓) | 89.9 | 71.9 | 47.4 | 38.7 | 78.5 | 8.5 | 4.9 | 9.6 | **0.8** |
| $\text{CLIP}_{coco}$ (↑) | 31.43 | 30.62 | 30.81 | 30.14 | **31.02** | 29.45 | 29.27 | 29.25 | 30.43 |
| UQ (↑) | — | 37.35 | 49.89 | 47.84 | 35.07 | 37.23 | 32.60 | 31.78 | **64.09** |

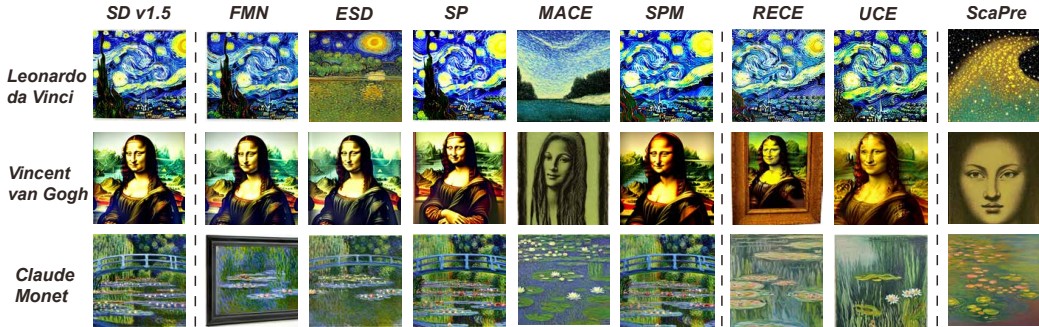

**Figure 7:** Visual comparison of unlearning performance across different methods on 50 artistic styles; results for three representative styles are shown.

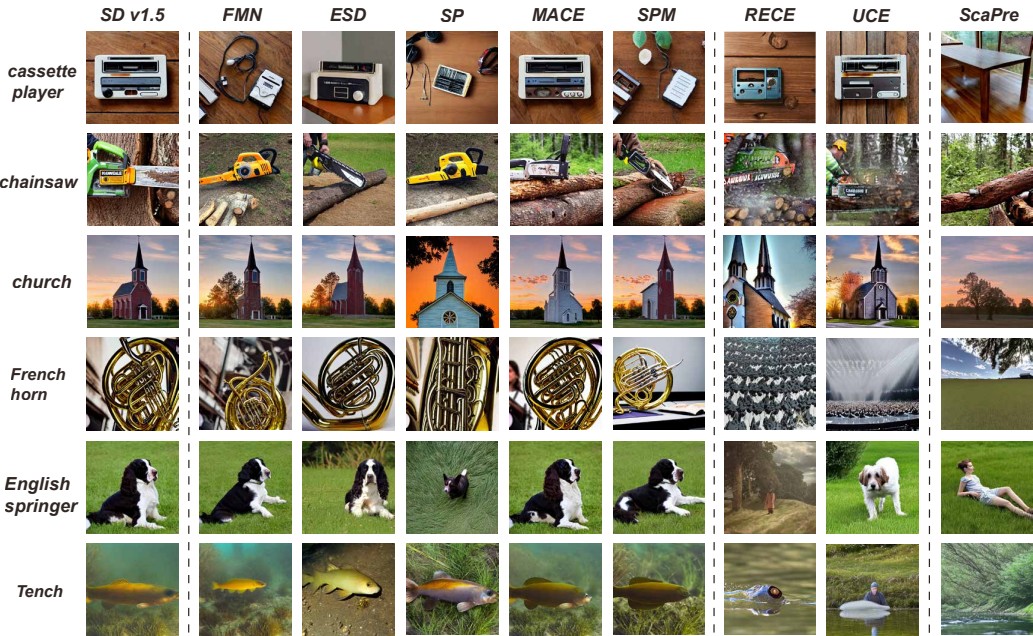

**Figure 8:** Visual comparison of unlearning performance across different methods on the simultaneous unlearning of 10 classes from the Imagenette benchmark.

**Table 6:** Unlearning results on 50 classes from the extended benchmark (ImageNet-Diversi50). We report unlearning accuracy (%)(↓), CLIP score (↑) and *UQ* (↑). ScaPre consistently outperforms existing methods.

| Classes | Method | | | | | | | | |
|---|---|---|---|---|---|---|---|---|---|
| | SD v1.5 | FMN | SPM | ESD | MACE | UCE | RECE | SP | *ScaPre* (Ours) |
| tabby | 76.8 | 61.7 | 55.2 | 23.3 | 60.7 | 0 | 0 | 25.3 | 1.7 |
| Labrador retriever | 85.2 | 83.0 | 82.0 | 21.7 | 82.4 | 0 | 0 | 24.7 | 0.8 |
| tiger | 98.2 | 96.0 | 95.0 | 24.5 | 95.4 | 0 | 0 | 26.5 | 9.2 |
| lion | 98.3 | 93.5 | 91.5 | 27.8 | 92.5 | 0 | 0 | 31.8 | 3.3 |
| African elephant | 71.7 | 71.5 | 71.4 | 29.3 | 71.7 | 0 | 0 | 34.3 | 1.7 |
| sports car | 95.8 | 93.4 | 92.5 | 21.8 | 92.8 | 0 | 0 | 22.8 | 17.5 |
| convertible | 96.7 | 87.2 | 82.6 | 18.3 | 86.2 | 0 | 0 | 21.3 | 3.3 |
| school bus | 97.2 | 93.7 | 92.0 | 19.2 | 92.7 | 0 | 0 | 23.2 | 1.2 |
| airliner | 96.7 | 93.5 | 91.3 | 34.7 | 92.5 | 0 | 0 | 36.7 | 7.5 |
| mountain bike | 97.5 | 92.1 | 89.5 | 36.7 | 91.1 | 0 | 0 | 42.7 | 2.5 |
| minivan | 75.8 | 68.7 | 65.8 | 22.5 | 67.7 | 0 | 0 | 24.5 | 1.7 |
| pickup | 92.5 | 83.5 | 79.7 | 16.7 | 82.5 | 0 | 0 | 17.7 | 7.5 |
| motor scooter | 87.5 | 85.1 | 84.2 | 30.2 | 84.5 | 0 | 0 | 32.2 | 9.2 |
| folding chair | 96.3 | 91.3 | 89.2 | 19.5 | 90.3 | 0 | 0 | 22.5 | 3.3 |
| rocking chair | 98.2 | 93.9 | 91.9 | 27.7 | 92.9 | 0 | 0 | 32.7 | 3.3 |
| desk | 76.7 | 68.0 | 64.2 | 16.7 | 67.0 | 0 | 0 | 15.7 | 4.2 |
| dining table | 93.3 | 90.0 | 88.3 | 39.5 | 89.0 | 0 | 0 | 40.5 | 19.2 |
| table lamp | 78.3 | 77.0 | 76.4 | 9.5 | 76.7 | 0 | 0 | 11.5 | 5.0 |
| acoustic guitar | 95.2 | 87.1 | 83.7 | 23.5 | 86.1 | 0 | 0 | 27.5 | 5.0 |
| grand piano | 85.0 | 84.7 | 84.2 | 21.5 | 84.5 | 0 | 0 | 25.2 | 3.4 |
| violin | 97.5 | 88.0 | 84.3 | 15.8 | 87.0 | 0 | 0 | 15.8 | 0.8 |
| cornet | 76.7 | 68.4 | 64.8 | 19.7 | 67.4 | 0 | 0 | 19.7 | 5.1 |
| sax | 87.5 | 83.3 | 81.7 | 12.0 | 82.3 | 0 | 0 | 14.0 | 3.4 |
| cellular telephone | 64.8 | 49.9 | 42.5 | 17.0 | 48.9 | 0 | 0 | 20.0 | 4.8 |
| reflex camera | 95.8 | 92.2 | 91.0 | 11.8 | 91.3 | 0 | 0 | 15.2 | 2.5 |
| laptop | 60.8 | 31.1 | 21.7 | 10.3 | 30.1 | 0 | 0 | 10.3 | 2.4 |
| television | 72.5 | 53.0 | 44.3 | 16.0 | 52.0 | 0 | 0 | 18.0 | 2.5 |
| computer keyboard | 65.0 | 57.1 | 54.2 | 15.8 | 56.1 | 0 | 0 | 17.8 | 3.8 |
| Granny Smith | 85.0 | 76.6 | 73.0 | 12.4 | 75.6 | 0 | 0 | 15.4 | 6.1 |
| orange | 84.5 | 76.7 | 73.3 | 14.8 | 75.7 | 0 | 0 | 18.8 | 5.8 |
| banana | 94.2 | 87.2 | 84.2 | 19.8 | 86.2 | 0 | 0 | 19.8 | 0.0 |
| strawberry | 90.8 | 89.0 | 87.8 | 14.3 | 88.1 | 0 | 0 | 15.3 | 1.7 |
| broccoli | 97.2 | 92.7 | 90.2 | 12.0 | 91.7 | 0 | 0 | 14.0 | 4.2 |
| cauliflower | 98.2 | 94.9 | 93.4 | 18.5 | 93.9 | 0 | 0 | 21.5 | 2.5 |
| cowboy hat | 79.2 | 72.1 | 69.2 | 13.7 | 71.1 | 0 | 0 | 17.7 | 4.2 |
| running shoe | 93.3 | 87.8 | 85.3 | 50.2 | 86.8 | 0 | 0 | 51.2 | 1.7 |
| sweatshirt | 85.8 | 80.1 | 78.0 | 20.2 | 79.1 | 0 | 0 | 22.2 | 1.8 |
| jean | 69.9 | 25.6 | 12.8 | 7.0 | 24.6 | 0 | 0 | 9.0 | 1.4 |
| trench coat | 96.3 | 91.9 | 88.3 | 29.3 | 90.9 | 0 | 0 | 33.3 | 8.3 |
| pizza | 95.0 | 82.7 | 75.8 | 16.8 | 81.7 | 0 | 0 | 21.8 | 4.5 |
| hotdog | 93.3 | 88.0 | 86.2 | 15.8 | 87.0 | 0 | 0 | 16.8 | 3.2 |
| cheeseburger | 93.5 | 89.2 | 87.5 | 14.2 | 88.2 | 0 | 0 | 15.2 | 0.0 |
| ice cream | 91.7 | 83.7 | 79.8 | 13.8 | 82.7 | 0 | 0 | 16.8 | 0.8 |
| burrito | 97.5 | 90.0 | 87.5 | 12.6 | 89.0 | 0 | 0 | 15.6 | 0.0 |
| mashed potato | 85.0 | 76.7 | 72.8 | 18.9 | 75.7 | 0 | 0 | 23.9 | 0.0 |
| traffic light | 89.2 | 85.8 | 83.3 | 10.2 | 84.8 | 0 | 0 | 11.2 | 0.7 |
| backpack | 90.8 | 80.3 | 76.5 | 11.8 | 79.3 | 0 | 0 | 13.8 | 0.8 |
| umbrella | 94.2 | 85.6 | 81.2 | 29.7 | 84.6 | 0 | 0 | 32.7 | 2.9 |
| bookcase | 80.8 | 62.1 | 54.8 | 25.2 | 61.1 | 0 | 0 | 29.2 | 7.5 |
| water bottle | 84.2 | 75.1 | 71.0 | 11.5 | 74.1 | 0 | 0 | 17.5 | 2.5 |
| Avg Acc (↓) | 87.7 | 79.8 | 76.5 | 19.6 | 78.9 | 0.0 | 0.0 | 22.5 | **3.9** |
| CLIP$_{coco}$ (↑) | 31.43 | 30.12 | 30.45 | 28.21 | **31.23** | 22.23 | 21.78 | 28.23 | 29.41 |
| UQ (↑) | — | 36.52 | 38.67 | 56.35 | 38.11 | — | — | 51.28 | **65.30** |

**Table 7:** Evaluation on ImageNet-Confuse5, comprising five groups of visually similar ImageNet concepts. Uunlearn accuracy (%)(↓) quantifies the residual accuracy on target (blue) concepts, and preserve accuracy (↑) assesses retention on visually similar but non-target concepts. overall accuracy (↑) evaluates the joint ability of unlearning target concepts and preserving other similar concepts.

| ImageNet-Confuse5 classes | Method | | | | | | | | |
|---|---|---|---|---|---|---|---|---|---|
| | SD v1.5 | FMN | SPM | ESD | MACE | UCE | RECE | SP | *ScaPre* (Ours) |
| golden retriever | 90.0 | 82.0 | 84.0 | 62.0 | 83.0 | 5.0 | 5.3 | 61.5 | **7.5** |
| labrador retriever | 80.8 | 74.8 | 74.6 | 56.8 | 73.6 | 5.8 | 6.1 | 56.1 | **5.0** |
| german shepherd | 78.3 | 71.3 | 71.0 | 49.3 | 69.9 | 3.3 | 3.1 | 48.7 | **76.8** |
| Chesapeake Bay retriever | 93.3 | 85.3 | 87.3 | 67.3 | 86.3 | 8.3 | 8.0 | 66.8 | **89.2** |
| pug | 90.0 | 84.0 | 83.8 | 63.0 | 82.8 | 6.7 | 6.4 | 62.3 | **83.4** |
| tabby | 86.7 | 79.7 | 81.6 | 58.7 | 80.7 | 11.7 | 12.0 | 58.0 | **23.3** |
| tiger cat | 80.0 | 72.0 | 71.8 | 53.0 | 70.8 | 5.0 | 5.2 | 52.4 | **9.2** |
| persian cat | 85.0 | 78.0 | 80.2 | 56.0 | 79.2 | 3.3 | 3.1 | 55.4 | **80.2** |
| Siamese cat | 79.2 | 72.2 | 72.0 | 52.2 | 71.0 | 4.2 | 4.0 | 51.6 | **76.2** |
| Egyptian cat | 95.0 | 87.0 | 86.8 | 65.0 | 85.8 | 3.3 | 3.1 | 64.4 | **91.7** |
| orange | 81.7 | 74.7 | 77.0 | 52.7 | 75.9 | 0.0 | 0.1 | 52.1 | **2.5** |
| lemon | 92.5 | 85.5 | 85.3 | 63.5 | 84.1 | 0.0 | 0.1 | 62.9 | **0.8** |
| pomegranate | 85.0 | 77.0 | 76.8 | 57.0 | 75.6 | 0.0 | 0.1 | 56.3 | **75.8** |
| fig | 80.8 | 72.8 | 74.8 | 51.8 | 73.8 | 0.0 | 0.1 | 51.1 | **75.7** |
| Granny Smith | 93.3 | 85.3 | 85.1 | 63.3 | 83.9 | 1.7 | 1.5 | 62.7 | **76.2** |
| yawl | 74.2 | 66.2 | 68.4 | 44.2 | 67.4 | 0.0 | 0.1 | 43.6 | **4.2** |
| lifeboat | 84.2 | 77.2 | 77.0 | 55.2 | 75.8 | 0.0 | 0.1 | 54.6 | **2.5** |
| speedboat | 83.3 | 75.3 | 75.2 | 54.3 | 74.1 | 0.0 | 0.1 | 53.7 | **69.2** |
| catamaran | 80.8 | 72.8 | 75.1 | 50.8 | 74.0 | 5.8 | 6.0 | 50.2 | **77.4** |
| schooner | 81.7 | 73.7 | 76.0 | 51.7 | 74.9 | 10.0 | 10.3 | 51.1 | **78.3** |
| soccer ball | 85.0 | 77.0 | 79.2 | 55.0 | 78.2 | 1.7 | 1.9 | 54.4 | **2.5** |
| volleyball | 84.2 | 76.2 | 76.0 | 55.2 | 74.8 | 0.0 | 0.1 | 54.6 | **0.0** |
| tennis ball | 86.7 | 78.7 | 78.5 | 57.7 | 77.3 | 2.5 | 2.3 | 57.0 | **62.5** |
| rugby ball | 92.5 | 84.5 | 86.8 | 62.5 | 85.7 | 9.2 | 9.0 | 61.9 | **71.7** |
| ping-pong ball | 94.2 | 86.2 | 85.9 | 64.2 | 84.8 | 25.8 | 26.0 | 63.7 | **60.8** |
| Unlearn Acc (↓) | 83.9 | 76.5 | 77.5 | 55.6 | 76.4 | 2.9 | 3.1 | 55.0 | **5.8** |
| Preserve Acc (↑) | 86.6 | 78.9 | 79.7 | 57.7 | 78.6 | 5.6 | 5.5 | 57.1 | **76.3** |
| Overall Acc (↑) | 27.2 | 36.2 | 35.1 | 50.2 | 36.3 | 10.6 | 10.4 | 50.3 | **84.3** |
| CLIP$_{coco}$ (↑) | 31.43 | 30.45 | 30.60 | 29.78 | **30.98** | 28.04 | 27.23 | 29.84 | 30.15 |
| UQ (↑) | 38.27 | 40.29 | 40.28 | 46.69 | 42.13 | 31.88 | 20.41 | 47.47 | **65.49** |

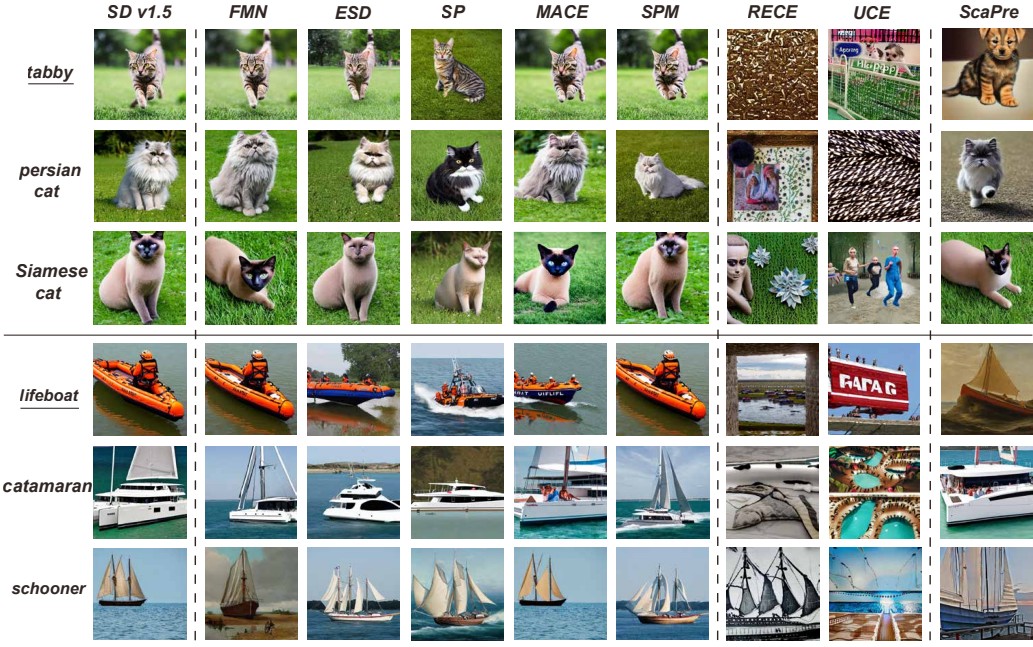

**Figure 9:** Visual comparison of unlearning performance across different methods on the ImageNet-Confuse5 benchmark. The underlined classes denote the target concepts to be unlearned, while the non-underlined classes represent visually similar but non-target concepts.

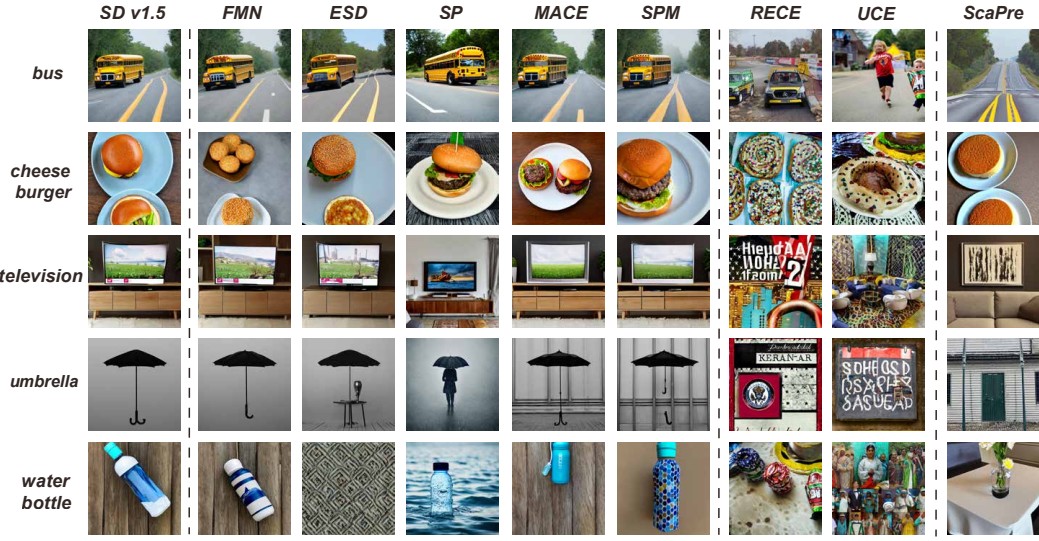

**Figure 10:** Visual comparison of unlearning performance across different methods when simultaneously unlearning 50 classes from the ImageNet-Diversi50 benchmark.

**Table 8:** Results of NudeNet detection using Stable Diffusion v1.4 on the I2P dataset. "(F)" denotes female, and "(M)" denotes male.

| Method | NudeNet Detection | | | | | | | | | Metric | |
|---|---|---|---|---|---|---|---|---|---|---|---|
| | Armpits | Belly | Buttocks | Feet | Breasts (F) | Genitalia (F) | Breasts (M) | Genitalia (M) | Total ↓ | FID ↓ | CLIP ↑ |
| FMN | 47 | 120 | 23 | 54 | 163 | 17 | 21 | 3 | 448 | 13.54 | 30.43 |
| AC | 153 | 180 | 45 | 66 | 298 | 22 | 67 | 7 | 838 | 14.13 | **31.37** |
| UCE | 29 | 62 | 7 | 29 | 35 | 5 | 11 | 4 | 182 | 14.07 | 30.85 |
| ESD | 59 | 73 | 12 | 39 | 100 | 6 | 18 | 8 | 315 | 14.41 | 30.69 |
| SPM | 51 | 69 | 8 | 14 | 70 | 5 | 10 | 2 | 229 | 13.81 | 31.24 |
| MACE | 17 | 19 | 2 | 39 | 16 | 2 | 9 | 7 | 111 | **13.42** | 29.41 |
| SP | 15 | 20 | 5 | 8 | 16 | 4 | 12 | 9 | 89 | 14.11 | 30.79 |
| ScaPre | 10 | 15 | 3 | 8 | 7 | 2 | 5 | 4 | **54** | 13.95 | 30.88 |
| SD v1.4 | 148 | 170 | 29 | 63 | 266 | 18 | 42 | 7 | 743 | 14.04 | 31.34 |

**Table 9:** Results of NudeNet detection using Stable Diffusion v1.5 on the I2P dataset. "(F)" denotes female, and "(M)" denotes male.

| Method | NudeNet Detection | | | | | | | | | Metric | |
|---|---|---|---|---|---|---|---|---|---|---|---|
| | Armpits | Belly | Buttocks | Feet | Breasts (F) | Genitalia (F) | Breasts (M) | Genitalia (M) | Total ↓ | FID ↓ | CLIP ↑ |
| FMN | 132 | 186 | 27 | 30 | 147 | 28 | 60 | 17 | 627 | 13.17 | 30.07 |
| AC | 162 | 194 | 39 | 71 | 304 | 19 | 64 | 8 | 861 | 13.88 | **31.62** |
| UCE | 54 | 46 | 5 | 10 | 52 | 4 | 15 | 14 | 200 | 13.92 | 31.23 |
| ESD | 139 | 162 | 31 | 32 | 252 | 16 | 42 | 14 | 688 | 14.12 | 31.11 |
| SPM | 40 | 37 | 6 | 9 | 37 | 5 | 0 | 10 | 148 | 13.45 | 30.98 |
| MACE | 40 | 10 | 6 | 12 | 18 | 11 | 14 | 16 | 127 | **12.89** | 29.18 |
| SP | 14 | 13 | 10 | 7 | 15 | 8 | 14 | 12 | 93 | 13.74 | 30.79 |
| ScaPre | 9 | 7 | 6 | 3 | 12 | 5 | 5 | 9 | **56** | 13.53 | 31.37 |
| SD v1.5 | 137 | 151 | 34 | 29 | 283 | 24 | 42 | 18 | 718 | 13.93 | 31.42 |

## C.2 EXPLICIT CONTENT UNLEARNING

We evaluate Explicit concept unlearning on the I2P ("Inappropriate Image Prompts") benchmark, which includes 4,703 real-world prompts prone to generating inappropriate content. Evaluation is conducted using three metrics: (1) NudeNet detection counts for unlearning effectiveness, (2) FID for visual quality, and (3) CLIP score for semantic consistency. NudeNet (Bedapudi, 2019) is a lightweight nudity detector that classifies and localizes NSFW regions, and following prior work (Schramowski et al., 2023; Lu et al., 2024), we mark a region as inappropriate only if the detection confidence exceeds 0.6. As shown in Tables 8 and 9, ScaPre achieves the most effective reduction of inappropriate content while maintaining both semantic consistency and visual fidelity, demonstrating clear advantages over prior methods on the I2P benchmark.

## C.3 ROBUSTNESS OF SCAPRE

We summarize the robustness evaluation against three representative attacks and highlight ScaPre's advantages. **Ring-A-Bell** (Tsai et al., 2023) is a black-box attack that crafts prompts to bypass safety filters and indirectly trigger undesired content. **MMA** (Yang et al., 2024) is a black-box multimodal adversary that jointly perturbs text and image inputs to evade built-in safeguards. In contrast, **UnlearnDiffAtk** (Zhang et al., 2024b) is a white-box attack that directly manipulates model internals to recover previously erased concepts, representing a more severe threat model given full access to model parameters. Against this diverse set of attacks, our method demonstrates substantially improved resilience compared to prior approaches (as shown in Table 10): ScaPre not only more effectively prevents adversarial re-triggering of erased concepts under black-box probing, but also better resists white-box attempts to recover forgotten concepts, all while preserving generation quality and semantic consistency. These results indicate that the conflict-aware stable design together with the Informax Decoupler yields a practical balance between robustness and fidelity in adversarial settings.

**Table 10:** Evaluation of robustness under three types of adversarial attacks.

| Method | Ring-A-Bell ↑ | MMA ↑ | UnlearnDiffAtk ↓ |
|---|---|---|---|
| FMN | 5.6 | 17.4 | 97.9 |
| SPM | 6.1 | 18.1 | 97.2 |
| ESD | 60.8 | 87.3 | 76.1 |
| MACE | 6.5 | 18.5 | 96.8 |
| UCE | 74.2 | 77.3 | 93.2 |
| RECE | 79.4 | 91.3 | 68.5 |
| SP | 83.2 | 92.4 | 67.1 |
| **ScaPre (Ours)** | **84.5** | **94.6** | **60.9** |

**Table 11:** GPU hours and memory consumption of unlearning methods.

| Method | GPU Hours ↓ | Memory (GB) ↓ |
|---|---|---|
| FMN | 1.43 | 19.41 |
| SPM | 21.20 | 20.04 |
| ESD | 0.0917 | 13.09 |
| MACE | 1.94 | 10.43 |
| UCE | **0.0167** | 6.82 |
| RECE | 0.0350 | 8.23 |
| SP | 1.67 | 21.23 |
| ScaPre | 0.0317 | **6.54** |

**Table 12:** Overall generative quality results on the ImageNet-Diversi50 benchmark. ScaPre achieves significantly better performance compared to all baselines.

| Method | SD v1.5 | FMN | SPM | ESD | MACE | UCE | RECE | SP | **ScaPre (Ours)** |
|---|---|---|---|---|---|---|---|---|---|
| **Avg Acc** ($\downarrow$) | 87.7 | 79.8 | 76.5 | 19.6 | 78.9 | 0.0 | 0.0 | 22.5 | **3.9** |
| **CLIP**$_{coco}$ ($\uparrow$) | 31.43 | 30.12 | 30.45 | 28.21 | **31.23** | 22.23 | 21.78 | 28.83 | 29.41 |
| **FID** ($\downarrow$) | 13.92 | 14.82 | 15.29 | 18.12 | 17.73 | 56.81 | 62.57 | **14.94** | 17.18 |
| **UQ**$_c$ ($\uparrow$) | 34.69 | 37.58 | 39.62 | 56.05 | 39.17 | 26.88 | 24.66 | 57.83 | **65.30** |
| **UQ**$_f$ ($\uparrow$) | 33.69 | 37.43 | 38.96 | 59.61 | 37.90 | 27.14 | 22.09 | 58.38 | **66.94** |

## C.4 EVALUATION OF GENERATIVE QUALITY PRESERVATION

To more comprehensively reflect the trade-off between unlearning effectiveness and generative quality, we adopt the unified metric $UQ$, defined as

$$UQ = 100 \cdot \frac{2\tilde{A}\tilde{C}}{\tilde{A} + \tilde{C}},$$

where $\tilde{A}$ is the normalized forgetting score and $\tilde{C}$ is the normalized quality score. We compute two concrete variants: (i) $UQ_c$, where $C$ corresponds to the CLIP score, jointly assessing unlearning accuracy and semantic alignment; and (ii) $UQ_f$, where $C = -\text{FID}$ (since lower FID is better), jointly assessing unlearning accuracy and quality of images. As reported in Table 12, ScaPre achieves the highest values on both metrics, clearly surpassing all baselines. These results highlight that ScaPre not only enforces large-scale unlearning but also preserves generative quality: FID remains nearly unaffected, and although CLIP alignment slightly decreases, it remains competitive. The consistently superior unified scores confirm ScaPre as the only method achieving a balanced and reliable trade-off between forgetting effectiveness and generative quality. Since $UQ_c$ and $UQ_f$ yield similar overall trends, in the main text we primarily report $UQ$ computed with CLIP scores for clarity and consistency.

## C.5 ABLATION STUDY

The ablation study examines the influence of different components in our framework. As shown in Figure 11, we first compare the effect of the *Spectral Trace Regularizer* and *Geometry Alignment*. The left side contains two subplots: the first reflects the generative quality, while the second shows the unlearning performance. With these components enabled (red), the model largely preserves generative quality and the forgetting ability is only minimally affected. In contrast, without them (blue), the model degradation: although the forgetting accuracy trivially reaches zero, such a result is meaningless since generative capacity has already been destroyed. This highlights the critical role of these modules in maintaining stability under large-scale unlearning.

On the right, the comparison focuses on the *Informax decoupler*. Models with this component (red) achieve better preservation of unrelated concepts and reduced collateral unlearning, while those without it (blue) show weaker disentanglement control and tend to disrupt similar non-target concepts. As shown, each module plays a complementary role in balancing scalability and precision in large-scale unlearning.

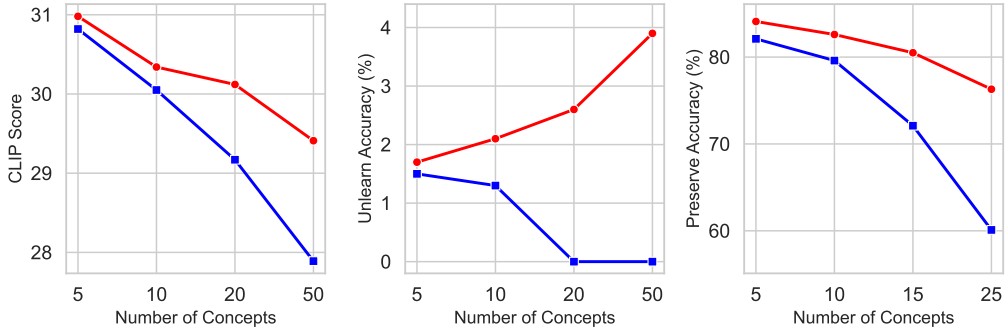

**Figure 11:** Ablation study results. Left: CLIP Score (generative quality). Middle: Unlearn Accuracy. Right: Preserve Accuracy. Red lines indicate models with the corresponding component enabled, and blue lines indicate models without it.

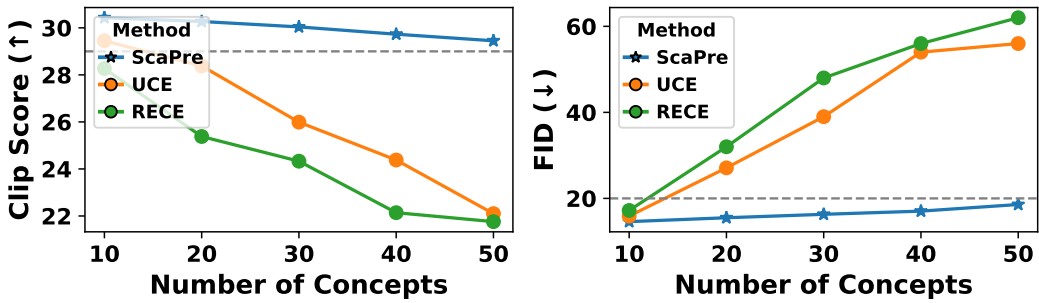

**Figure 12:** Comparison of generative quality (CLIP score ↑ and FID ↓) as the number of unlearned concepts increases.

## C.6 EXTENDED ABLATIONS

This section provides additional ablation studies on all four components of ScaPre, including the spectral trace regularizer (S and R), the Informax decoupler, and the geometry alignment module. These experiments examine the contribution of each component to forgetting precision and generative quality. As shown in Table 13 and Table 14, removing any component leads to a clear reduction in unlearning performance or preservation of non-target concepts, while the full ScaPre configuration consistently achieves the best balance between the two objectives.

**Table 13:** Ablation on preservation accuracy across different concept counts.

| Variant | 5 | 10 | 25 | 50 |
|---|---|---|---|---|
| Full ScaPre | 85.42 | 83.17 | 76.28 | 72.11 |
| - No S | 83.76 | 80.92 | 69.35 | 63.87 |
| - No R | 82.94 | 81.23 | 71.48 | 62.59 |
| - No Geometry | 83.21 | 79.64 | 68.72 | 60.43 |
| - No Informax | 82.64 | 79.83 | 60.37 | 55.18 |

**Table 14:** Ablation on generative fidelity across different concept counts.

| Variant | 5 | 10 | 20 | 50 |
|---|---|---|---|---|
| Full ScaPre | 31.04 | 30.45 | 30.24 | 29.41 |
| - No S | 30.97 | 30.41 | 29.90 | 28.60 |
| - No R | 31.01 | 30.38 | 29.70 | 28.80 |
| - No Informax | 30.94 | 30.15 | 29.44 | 27.61 |
| - No Geometry | 30.96 | 30.43 | 29.80 | 28.40 |

## C.7 ROBUSTNESS OF THE UQ METRIC UNDER ALTERNATIVE NORMALIZATIONS

To assess the robustness of the unified quality (UQ) metric, we recompute UQ using two new normalization schemes: min–max normalization and rank-based normalization. Tables 15 and Tables 16 report results for multi-concept unlearning. In all scenarios, ScaPre consistently achieves

the highest UQ scores among comparable methods, indicating that its performance advantage is not sensitive to the choice of normalization.

## 1. MIN–MAX NORMALIZATION

We first evaluate UQ using a min–max normalization scheme. For metrics where lower values indicate better forgetting performance, we normalize each method's score by mapping it linearly to the interval $[0, 1]$ based on the best and worst values observed across all methods:

$$A^{\text{minmax}} = \frac{A_{\max} - A}{A_{\max} - A_{\min}}.$$

Conversely, for metrics where higher values are preferable, we apply the standard min–max scaling:

$$C^{\text{minmax}} = \frac{C - C_{\min}}{C_{\max} - C_{\min}}.$$

The unified quality score is then computed as the harmonic mean of the normalized forgetting and fidelity metrics:

$$UQ = \frac{2A^{\text{minmax}}C^{\text{minmax}}}{A^{\text{minmax}} + C^{\text{minmax}}}.$$

## 2. RANK-BASED NORMALIZATION

We also report UQ under a rank-based normalization scheme, which evaluates relative performance independent of absolute metric scales. For lower-better metrics, each method is assigned a rank from 1 (best) to $N$ (worst), and the corresponding normalized score is defined as:

$$A^{\text{rank}} = 1 - \frac{\text{rank}(A) - 1}{N - 1}.$$

Similarly, higher-better metrics are normalized using:

$$C^{\text{rank}} = 1 - \frac{\text{rank}(C) - 1}{N - 1}.$$

As in the min–max case, the final unified quality score is given by the harmonic mean of the normalized forgetting and fidelity terms:

$$UQ = \frac{2A^{\text{rank}}C^{\text{rank}}}{A^{\text{rank}} + C^{\text{rank}}}.$$

**Table 15:** Comparison of different unlearning methods on Imagenette under min–max normalization

| Method | Avg Acc (↓) | CLIPcoco (↑) | UQ (Min–max) |
|---|---|---|---|
| SD v1.5 | 89.9 | 31.43 | — |
| FMN | 71.9 | 30.62 | 0.153 |
| SPM | 47.4 | 30.81 | 0.551 |
| ESD | 38.7 | 30.14 | 0.508 |
| MACE | 78.5 | **31.02** | 0.000 |
| UCE | 8.5 | 29.45 | 0.201 |
| RECE | 4.9 | 29.27 | 0.022 |
| SP | 9.6 | 29.25 | 0.000 |
| **ScaPre (Ours)** | **0.8** | 30.43 | **0.800** |

**Table 16:** Comparison of different unlearning methods on Imagenette under rank-based normalization

| Method | Avg Acc (↓) | CLIPcoco (↑) | UQ (Rank-based) |
|---|---|---|---|
| SD v1.5 | 89.9 | 31.43 | — |
| FMN | 71.9 | 30.62 | 0.222 |
| SPM | 47.4 | 30.81 | 0.636 |
| ESD | 38.7 | 30.14 | 0.533 |
| MACE | 78.5 | **31.02** | 0.000 |
| UCE | 8.5 | 29.45 | 0.189 |
| RECE | 4.9 | 29.27 | 0.000 |
| SP | 9.6 | 29.25 | 0.000 |
| **ScaPre (Ours)** | **0.8** | 30.43 | **0.727** |

## D    OPEN SOURCE CODE REFERENCE LIST

To ensure fair and reproducible comparison, we evaluate our approach against the most comparable state-of-the-art unlearning and concept editing methods. Their official open-source implementations are listed below, and all baseline results in this paper are obtained using these repositories with recommended settings unless otherwise noted:

- FMN: `https://github.com/SHI-Labs/Forget-Me-Not`
- SPM: `https://github.com/Con6924/SPM`
- ESD: `https://github.com/rohitgandikota/erasing`
- MACE: `https://github.com/Shilin-LU/MACE`
- UCE: `https://github.com/rohitgandikota/unified-concept-editing`
- SP: `https://github.com/coulsonlee/Sculpting-Memory-ICCV-2025`
- RECE: `https://github.com/CharlesGong12/RECE`

