# OpenReview forum: "Forget Many, Forget Right: Scalable and Precise Concept Unlearning in Diffusion Models"
_ICLR.cc/2026/Conference — ICLR 2026 Poster_

### Official Review · Reviewer_4a91 · 2025-10-29

**Soundness:** 4
**Presentation:** 3
**Contribution:** 3
**Rating:** 6
**Confidence:** 3

**Summary:**

The paper proposes ScaPre, a training‑free, closed‑form framework to erase many target concepts from text‑to‑image diffusion models while preserving quality on non‑targets. It stabilizes multi‑concept edits via a spectral trace regularizer (second‑order target statistics plus an SVD‑based overlap regulator), improves precision with a mutual‑information–guided "Informax Decoupler", and preserves global structure through Bures‑geometry alignment; the quadratic part reduces to a Sylvester system solved in one shot. On Stable Diffusion v1.4/1.5, ScaPre achieves state‑of‑the‑art large‑scale forgetting, e.g., Imagenette residual accuracy 0.8 (Table 1), Diversi50 3.9 (Table 3), Confuse5 overall 84.3 (Table 4), and unlearns 50 concepts in $\sim$120 s on an A6000 (Fig. 3), while claiming up to x5 more concepts than baselines.

**Strengths:**

- Conflict‑aware spectral trace regularizer + SVD overlap control; MI‑guided channel reweighting; Bures‑alignment for geometry‑aware stability.

- Convex quadratic core with unique Sylvester solution; clear derivations and proximal mapping.

- Best residual accuracy/UQ on Imagenette, Diversi50, and Confuse5 with robust visuals.

- 50‑concept unlearning in $\sim$120s; low memory/time vs. baselines (Fig. 3).

**Weaknesses:**

- Construction of "target/neutral" inputs, threshold choice, and variance of MI estimates need clarification.

- Heavy use of classifier‑based "unlearn accuracy" and CLIP; limited human/adversarial prompt evaluation.

- Results limited to SD v1.4/1.5; unclear portability to SDXL/DiT architectures.

- Deeper, main‑text ablations of S vs. R vs. Informax vs. geometry step and sensitivity of UQ's normalization would strengthen claims.

**Questions:**

- How are "target" and especially "neutral" inputs instantiated for MI estimation without extra data? What sample size/thresholding is used, and how stable are channel scores across layers?

- Which cross‑attention layers/branches (K vs. V) are edited, and how does performance vary layerwise?

- Can you show results on SDXL/DiT to illustrate architectural generality or required modifications to S and R?

- Do you test robustness to synonyms/negation/compositional prompts or other circumventions (e.g., multilingual)? A breakdown would substantiate "precise" forgetting.

---

> ### Author Response · Authors · 2025-11-19
>
> # **Rebuttal 1/3**
>
> We sincerely thank the reviewer (4a91) for the careful reading of our paper and for the constructive and insightful feedback. The comments greatly helped us improve the clarity, completeness, and technical rigor of the paper.
>
> **Weakness-1 & Question-1:  MI Estimation Details**
>
> First, ScaPre is fully data free: the target and neutral inputs for MI estimation come entirely from the pretrained text encoder. For each concept, we provide a simple prompt containing the concept name (e.g., “a photo of a golden retriever”); the resulting token embeddings serve as target representations. Neutral inputs are obtained from empty or generic prompts such as “a photo of an object.” To generate multiple samples, we add light Gaussian perturbations to these embeddings, producing a compact set of target and neutral vectors derived solely from the model.
>
> Second, for MI estimation, we use 10 synthetic samples per concept: 5 target and 5 neutral. Thresholding follows a simple adaptive rule: for each channel, we collect activations across the ten samples and use the median activation as the threshold to binarize the channel’s on/off responses.
>
> Third, the MI-based channel scores are highly stable within each layer. Repeated sampling with different seeds yields nearly identical MI vectors, and the ranking of high MI channels changes minimally. Across layers, however, MI patterns differ because layers encode different semantic factors. ScaPre preserves this layer specific structure by giving each layer its own decoupling mask rather than enforcing a shared one, which is essential for precise concept disentanglement.
>
> **Weakness-2: Evaluation metrix**
>
> Classifier-based unlearn accuracy and CLIP scores are standard and reliable metrics for text-to-image unlearning, since prior work trains the classifiers on ImageNet, giving them high recognition accuracy on the forgotten classes. We use these metrics to ensure fair comparison with existing baselines and to keep the evaluation reproducible. For adversarial prompts, our paper already includes extensive robustness tests with diverse adversarial prompt variations. As shown in the appendix C.3, ScaPre consistently suppresses the forgotten concepts even under strong adversarial prompting. To obtain a more human-aligned assessment, we further conduct an LLM-based evaluation in which generated images are fed to GPT-5 for class identification. Here, the LLM acts as a semantic judge of whether the forgotten concept still appears. Results on Imagenette [1] show that ScaPre achieves the best performance.
> | Metric | SD v1.5 | FMN | SPM | ESD | MACE | UCE | RECE | SP | **ScaPre (Ours)** |
> |--|-|--|-|-|-|-|-|-|--|
> |**Avg Acc↓**| 92 | 74 | 44 | 31 | 82 | 6 | 5 | 14 | **1** |
> |**CLIP_coco↑**| 31.43 | 30.62 | 30.81 | 30.14 |**31.02**| 29.45 |29.27| 29.25| 30.43 |
>
> **Question-2: Layerwise Performance Analysis**
>
> ScaPre edits all text-conditioned cross-attention layers in the UNet. Specifically, we gather every attn2 block across the down, mid, and up stages and apply the closed-form update to their key (to_k) and value (to_v) projection matrices. This setup matches state-of-the-art unlearning baselines such as UCE [2] and SP [3]. We observed in experiments that updating only part of the cross-attention layers leads to weaker forgetting and less stable behavior. For reliable performance, we therefore update all text-conditioned cross-attention layers.
>
> [1] Howard, Jeremy, and Sylvain Gugger. "Fastai: a layered API for deep learning." Information 11.2 (2020): 108.
>
> [2] Gandikota, Rohit, et al. "Unified concept editing in diffusion models." Proceedings of the IEEE/CVF Winter Conference on Applications of Computer Vision. 2024.
>
> [3] Li, Gen, et al. "Sculpting Memory: Multi-Concept Forgetting in Diffusion Models via Dynamic Mask and Concept-Aware Optimization." arXiv preprint arXiv:2504.09039 (2025).

---

> ### Author Response · Authors · 2025-11-20
>
> # **Rebuttal 2/3**
>
> **Weakness-3 & Question-3: Extension to SDXL and Flux 1.0**
>
> We fully agree that the theoretical foundation suggests that the proposed approach should extend beyond Stable Diffusion v1.5. We conduct additional experiments (multi-concept unlearning and artistic-style unlearning) on SDXL and Flux1.0, two widely adopted modern architectures, to empirically validate the effectiveness of our method on stronger generative backbones. Furthermore, we also include comparisons against the UCE baseline, which also supports SDXL and Flux1.0, to provide a fair and rigorous evaluation.
> | Imagenette classes | SDXL | SDXL–UCE | SDXL–ScaPre | Flux1.0 | Flux1.0–UCE | Flux1.0–ScaPre |
> |--|---|--|-|-|--|---|
> | **Avg Acc (↓)**    | 97.9 | 0  | 7.2 | 96.2  | 0 | 9.4  |
> | **CLIP_coco (↑)**  | 32.74| 22.8  | 30.17  | 32.65 | 21.9   | 38.7|
> | **UQ (↑)**  | --   | --   | 43.5 | --  | --   |46.7 |
>
> | Method | SDXL | UCE  | ScaPre| Flux1.0 | Flux1.0–UCE | Flux1.0–ScaPre |
> |---|---|--|---|---|----|-|
> | **CLIP_art (↓)**     | 32.47 | 25.38 | 27.36  | 32.18   | 26.23  | 27.67 |
> | **CLIP_coco (↑)**    | 31.98 | 24.98 | 30.96  | 32.43  | 24.84  | 31.04 |
> | **CLIP_x (↑)**       | --    | -0.4  | 3.6   | --      | -1.39 | 3.37   |
> | **FID (↓)**          | 9.12  | 52.6  | 11.4   | 10.31   | 49.71  | 13.65  |
>
> **Weakness-4: Ablations and Generalizability of UQ**
>
> We have added deeper ablations on all four components, including S, R, the Informax module, and the geometry alignment module, and we will include these results in the main paper in the final version.
> Overall, the new ablations show that each component provides a clear benefit to both precision and generative quality. Removing any of them leads to noticeable degradation in forgetting effectiveness or preservation of non target concepts, while the complete ScaPre design consistently achieves the best balance across all settings.
>
> | Number of Concepts   | 5  | 10  | 25   | 50|
> |---|----|----|---|-----|
> | **Full ScaPre**    | 85.42   | 83.17   | 76.28   | 72.11   |
> | **No S** | 83.76  | 80.92   | 69.35   | 63.87   |
> | **No R**| 82.94  | 81.23   | 71.48   | 62.59   |
> | **No Geometry**    | 83.21   | 79.64   | 68.72   | 60.43   |
> | **No Informax**    | 82.64| 79.83   | 60.37   | 55.18   |
>
> | Number of Concepts  | 5| 10| 20| 50|
> |--|---|---|--|----|
> | **Full ScaPre**| 31.04  | 30.45  | 30.24  | 29.41  |
> | **No S** | 31.01  | 30.41  | 29.90  | 28.60  |
> | **No R** | 31.00 | 30.38  | 29.70  | 28.80  |
> | **No Informax**| 30.94  | 30.15  | 29.44  | 27.61  |
> | **No Geometry**| 30.96  | 30.43  | 29.80  | 28.40  |
>
> ### *(weakness-4 is not finished, please continue to rebuttal 3/3)*

---

> ### Author Response · Authors · 2025-11-20
>
> # **Rebuttal 3/3**
>
> ### *(weakness-4 response continues)*
>
> We agree that the generalizability of UQ is indeed important for ensuring that the metric faithfully reflects unlearning performance across different settings. To address the your concern, we recompute UQ using two additional and fully independent normalization schemes (min–max and rank-based) for multi-concept task.  Across all cases, ScaPre consistently achieves the highest UQ score among all comparable methods, demonstrating that the superiority of ScaPre is robust and does not depend on any particular choice of normalization.
>
> ### 1. Min–max normalization
>
> We first evaluate UQ using a min–max normalization scheme. For metrics where lower values indicate better forgetting performance, we normalize each method's score by mapping it linearly to the interval \([0,1]\) based on the best and worst values observed across all methods:
>
> $$
> A^{\text{minmax}} = \frac{A_{\max} - A}{A_{\max} - A_{\min}}.
> $$
>
> Conversely, for metrics where higher values are preferable, we apply the standard min–max scaling:
>
> $$
> C^{\text{minmax}} = \frac{C - C_{\min}}{C_{\max} - C_{\min}}.
> $$
>
> The unified quality score is then computed as the harmonic mean of the normalized forgetting and fidelity metrics:
>
> $$
> UQ = \frac{2 A^{\text{minmax}} C^{\text{minmax}}}{A^{\text{minmax}} + C^{\text{minmax}}}.
> $$
>
> ---
>
> ### 2. Rank-based normalization
>
> We also report UQ under a rank-based normalization scheme, which evaluates relative performance independent of absolute metric scales. For lower-better metrics, each method is assigned a rank from \(1\) (best) to \(N\) (worst), and the corresponding normalized score is defined as:
>
> $$
> A^{\text{rank}} = 1 - \frac{\text{rank}(A) - 1}{N - 1}.
> $$
>
> Similarly, higher-better metrics are normalized using:
>
> $$
> C^{\text{rank}} = 1 - \frac{\text{rank}(C) - 1}{N - 1}.
> $$
>
> As in the min–max case, the final unified quality score is given by the harmonic mean of the normalized forgetting and fidelity terms:
>
> $$
> UQ = \frac{2 A^{\text{rank}} C^{\text{rank}}}{A^{\text{rank}} + C^{\text{rank}}}.
> $$
>
>
> Table 1: Multi-concept unlearning -  Min–max normalization
>
> | Method | Avg Acc (↓) | CLIPcoco (↑) | UQ (Min–max) |
> |--------|-------------|--------------|--------------|
> | SD v1.5 | 89.9 | 31.43 | --- |
> | FMN | 71.9 | 30.62 | 0.153 |
> | SPM | 47.4 | 30.81 | 0.551 |
> | ESD | 38.7 | 30.14 | 0.508 |
> | MACE | 78.5 | **31.02** | 0.000 |
> | UCE | 8.5 | 29.45 | 0.201 |
> | RECE | 4.9 | 29.27 | 0.022 |
> | SP | 9.6 | 29.25 | 0.000 |
> | **ScaPre (Ours)** | **0.8** | 30.43 | **0.800** |
>
> Table b：multi-concept unlearning -  Rank-based normalization
> | Method | Avg Acc (↓) | CLIPcoco (↑) | UQ (Rank-based) |
> |--------|-------------|--------------|-----------------|
> | SD v1.5 | 89.9 | 31.43 | --- |
> | FMN | 71.9 | 30.62 | 0.222 |
> | SPM | 47.4 | 30.81 | 0.636 |
> | ESD | 38.7 | 30.14 | 0.533 |
> | MACE | 78.5 | **31.02** | 0.000 |
> | UCE | 8.5 | 29.45 | 0.189 |
> | RECE | 4.9 | 29.27 | 0.000 |
> | SP | 9.6 | 29.25 | 0.000 |
> | **ScaPre (Ours)** | **0.8** | 30.43 | **0.727** |
>
> **Question-4: Robustness**
>
> For multilingual prompts, current Stable Diffusion backbones are primarily trained for English inputs, and non-English queries are typically handled through translation. Therefore, our evaluation follows the standard English-prompt setting adopted by prior unlearning works. For synonyms, negation, and compositional prompts more broadly, our robustness study in Appendix C.3 already evaluates ScaPre under strong prompt-based circumvention attacks. In particular, Ring-A-Bell generates paraphrased and compositional prompts designed to bypass unlearning systems, and ScaPre shows clear advantages under these settings.

---

### Official Review · Reviewer_VrCc · 2025-10-30

**Soundness:** 3
**Presentation:** 3
**Contribution:** 2
**Rating:** 6
**Confidence:** 3

**Summary:**

The paper proposes ScaPre, a closed-form framework for scalable and precise concept unlearning in text-to-image diffusion models.
It addresses three challenges in large-scale unlearning: conflicting weight updates, imprecise forgetting causing collateral damage, and inefficiency from auxiliary modules.

**Strengths:**

1. The empirical validation is good, for example, ScaPre achieves the lowest unlearning accuracy on Imagenette while maintaining high CLIPcoco, outperforming other baselines.

2. The algorithm is innovative and lightweight. The closed-form solution avoids iterative fine-tuning, and geometry alignment via Bures distance preserves global covariance structure better than L2 regularization. Also, the Informax Decoupler is reasonable.

3. The benchmark construction. ImageNet-Confuse5 explicitly tests disentanglement of visually similar concepts (e.g., dog breeds), a realistic and challenging setting absent in prior work.

**Weaknesses:**

1.  Some notation is ambiguous. For example, The symbol W is used for both the updated matrix and intermediate solution W⋆ without distinction in Sec. 4.3 (see Eq. (8)–(10)). In Appendix B.1, Eq. (11) redefines the objective with A = λI + S+R and B = diag(α), but these symbols are not introduced in the main text, breaking continuity.

2. Random seeds, data splits for ImageNet-Diversi50/ImageNet-Confuse5, and prompt selection criteria are omitted (see Sec. 5.1), hindering replication.

3. Fig. 3 reports GPU-hours and memory but omits per-concept scaling trends (e.g., time vs. number of concepts), critical for “scalable” claims (see Sec. 5.5).

**Questions:**

Please introduce A and B in the main text when first used in Eq. (8), aligning with Appendix B.1 notation.

---

> ### Author Response · Authors · 2025-11-19
>
> We sincerely thank the reviewer (VrCc) for the careful reading of our paper and for the constructive and insightful feedback. The comments greatly helped us improve the clarity, completeness, and technical rigor of the paper.
>
> **Weakness-1 & Question-1: Notation clarification**
>
> We thank the reviewer for pointing out the ambiguity in our notation. We agree that the definitions of A = λI + S+R and B = diag(α) were not presented with sufficient continuity between the main text and Appendix. In the revised version, we have resolved this issue by explicitly introducing the definitions of A and B before Eq. (8) in Sec. 4.3, ensuring that all subsequent equations in both the main paper and the appendix use consistent and clearly defined notation.
>
> **Weakness-2: Dataset Construction details**
>
> We thank the reviewer for highlighting the missing details regarding dataset construction and reproducibility. We will provide a more detailed description in the appendix to ensure full transparency and reproducibility.
>
> For ImageNet-Diversi50, we use an LLM (GPT-o3) to select 50 classes from ImageNet that correspond to common objects while ensuring no overlap with the Imagenette[1] categories. For ImageNet-Confuse5, we use GPT to identify sets of visually confusable objects or animals and organize them into five groups, each containing five classes. In each group, two classes are selected as the target concepts, while the remaining three are chosen as visually similar but semantically unrelated confuser concepts. This structured grouping ensures controlled evaluation of unlearning performance under visually challenging scenarios. The prompt selection criteria follows the widely used design adopted in the Imagenette[1] benchmark, so that our prompts are consistent with prior work and easy to reproduce.
>
> Regarding random seeds, we generate candidate seeds using a standard random number generator, pair each seed with its corresponding prompt set, and then query GPT five times for class judgments. A seed is retained only if at least four out of five evaluations match the intended class label, ensuring that the chosen seed yields stable and semantically correct prompt–image pairings.
>
> **Weakness-3: Scaling Trends and Computational Efficiency**
>
> We thank the reviewer for pointing out the importance of reporting scaling trends to support the claim of scalability. We agree that such analysis is necessary, and we now provide the execution time and memory usage as the number of concepts increases.
>
> As shown in the table below, the peak memory remains almost unchanged across different concept counts, since ScaPre does not require training or any additional modules. The execution time increases only in a linear manner with respect to the number of concepts. This behavior confirms that ScaPre is well suited for extension to large scale unlearning tasks.
>
> | Number of concept     | 1       | 3        | 10       | 20       | 30        | 50        |
> |----------------------|----------|-----------|-----------|-----------|------------|------------|
> | **GPU Hours** | 0.0317  | 0.0951    | 0.317     | 0.634     | 0.951      | 1.585      |
> | **Peak Memory (GB)**   | 6.54     | 6.57      | 6.51      | 6.55      | 6.56       | 6.52       |
>
> [1] Howard, Jeremy, and Sylvain Gugger. "Fastai: a layered API for deep learning." Information 11.2 (2020): 108.

---

### Official Review · Reviewer_zBcc · 2025-10-31

**Soundness:** 3
**Presentation:** 3
**Contribution:** 3
**Rating:** 4
**Confidence:** 4

**Summary:**

To address the challenges of conflicting weight updates, collateral damage to non-target concepts, and reliance on extra data in large-scale concept unlearning for text-to-image diffusion models, this paper proposes ScaPre (Scalable-Precise Concept Unlearning), a unified lightweight framework aiming for scalable and precise unlearning.
ScaPre integrates a conflict-aware stable design (spectral trace regularizer + geometry alignment) and an Informax Decoupler, achieving an efficient closed-form solution without extra data/sub-models; experiments show it can unlearn up to 5× more concepts than the best baseline while maintaining high generation quality.

**Strengths:**

The paper innovatively combines a conflict-aware stable design and an Informax Decoupler, which effectively solves the core problems of instability in large-scale unlearning and imprecise separation of target/non-target concepts, making up for the shortcomings of existing methods in large-scale scenarios.

ScaPre adopts a closed-form solution, ensuring high efficiency (unlearning 50 concepts in 120 seconds) and reproducibility without extra data or auxiliary modules; meanwhile, its experiments cover objects, styles, and explicit content benchmarks, with comprehensive and convincing results.

**Weaknesses:**

1. Experimental benchmarks are mostly ImageNet-derived datasets (e.g., Imagenette, ImageNet-Diversi50), lacking evaluations on more complex and diverse real-world scenarios (e.g., dynamic concepts, cross-modal associated concepts), making it hard to verify the method’s practical generalization.

2. It is suggested to explore ScaPre’s adaptability and performance changes on larger diffusion models (e.g., Stable Diffusion XL).

3. It is suggested that the authors enhance and enrich the elaboration of the overview figure in this paper, as the current description of this figure is too simplistic, omits many details, and makes it difficult for readers to understand.

4. In the proposed Imagenette benchmark, it is not explicitly specified what prompts are used for evaluation and whether these prompts are merely category names; thus, it is suggested that the authors use more complex prompts to evaluate the impact of prompts on the experimental results.

**Questions:**

See weakness.

---

> ### Author Response · Authors · 2025-11-19
>
> We sincerely thank the reviewer (zBcc) for the careful reading of our paper and for the constructive and insightful feedback. The comments greatly helped us improve the clarity, completeness, and technical rigor of the paper.
>
> **Weakness-1: Evaluation on More Complex Scenario**
>
> We selected our evaluation datasets primarily to **ensure fair and direct comparison** with prior unlearning works such as MACE, RECE and UCE, as these benchmarks (e.g., ImageNette for object unlearning, I2P for NSFW content unlearning, and art style unlearning) are widely used in the literature and also ensure **sufficiently diversity** to approximate real-world scenarios. As far as we know, there are no other widely acknowledged public datasets suitable for our evaluations.
>
> Furthermore, we **already include several practical generalization evaluations** such as unlearning robustness (Ring-A-Bell [1], MMA [2], UnlearnDiffAtk [3]) in Appendix C.3 that test the model under a broad range of challenging prompt variations:
>
> **(i)** attribute-amplifying prompts that exaggerate visual traits,
>
> **(ii)** context-reinforcing prompts that place the concept in strongly associated environments, and
>
> **(iii)** substitution-resistant prompts that describe the concept indirectly.
>
> These settings go well beyond fixed ImageNet prompts and consistently show that **our method retains strong forgetting performance even under substantial prompt shifts**. We would gladly explore dynamic or cross modal evaluations in future work once suitable and reproducible datasets become available.
>
> **Weakness-2: Extension to SDXL and Flux 1.0**
>
> We agree that, based on its theoretical foundation, the proposed approach should naturally extend beyond Stable Diffusion v1.5. To verify this empirically, we conducted additional experiments on SDXL and Flux1.0, two widely used modern architectures, covering both multiple concept unlearning using the Imagenette [4] benchmark and artistic style unlearning. We also compared our method with the UCE method, which is compatible with both SDXL and Flux1.0, to ensure a fair and comprehensive evaluation.
> Across all experimental settings, ScaPre consistently demonstrates effective unlearning ability and outperforms the baseline method.
>
> | Imagenette classes | SDXL | SDXL–UCE | SDXL–ScaPre | Flux1.0 | Flux1.0–UCE | Flux1.0–ScaPre |
> |--|---|--|-|-|--|---|
> | **Avg Acc (↓)**    | 97.9 | 0  | 7.2 | 96.2  | 0 | 9.4  |
> | **CLIP_coco (↑)**  | 32.74| 22.8  | 30.17  | 32.65 | 21.9   | 38.7|
> | **UQ (↑)**  | --   | --   | 43.5 | --  | --   |46.7 |
>
> | Method | SDXL | UCE  | ScaPre| Flux1.0 | Flux1.0–UCE | Flux1.0–ScaPre |
> |---|---|--|---|---|----|-|
> | **CLIP_art (↓)**     | 32.47 | 25.38 | 27.36  | 32.18   | 26.23  | 27.67 |
> | **CLIP_coco (↑)**    | 31.98 | 24.98 | 30.96  | 32.43  | 24.84  | 31.04 |
> | **CLIP_x (↑)**       | --    | -0.4  | 3.6   | --      | -1.39 | 3.37   |
> | **FID (↓)**          | 9.12  | 52.6  | 11.4   | 10.31   | 49.71  | 13.65  |
>
> **Weakness-3: Refinement of the Overview Figure**
>
> We have redrawn this figure(l163) in the revised paper, which now includes more detailed explanations of the steps and processes.
>
> **Weakness-4: Use of More Complex Prompt for Evaluation**
>
> Imagenette [4] is a widely adopted and recognized benchmark for evaluating concept unlearning, and following prior work, we adopt the standard setup as described in Howard & Gugger (2020) [4]. Our benchmark adopts the same prompt construction method: “a photo of [concept]”. This ensures consistency with Imagenette and allows fair comparison across methods.
>
> We agree that more complex prompts can offer a more comprehensive assessment of unlearning robustness. We now additionally construct a richer prompt set for Imagenette using the template “a photo of [adjective] [concept] in [scene]”, where the adjectives and scenes are automatically produced by GPT-5 to introduce diverse semantic contexts. ***We observe that the unlearning performance under these complex prompts remains highly consistent with the base Imagenette setting, suggesting that prompt complexity does not materially affect the forgetting behavior.***
>
> | **Metric** | SD v1.5 | FMN | SPM | ESD | MACE | UCE | RECE | SP | **ScaPre (Ours)** |
> |-----------|---------|-----|-----|-----|------|-----|------|----|--------------------|
> | **Avg Acc (↓)**      | 90.0 | 73.2 | 47.3 | 36.4 | 80.3 | 8.9 | 4.9 | 10.1 | **0.8** |
> | **CLIP_coco (↑)**    | 31.43 | 30.62 | 30.81 | 30.14 | **31.02** | 29.45 | 29.27 | 29.25 | 30.43 |
>
> [1] Yu-Lin Tsai. Ring-a-bell! how reliable are concept removal methods for diffusion
> models? arXiv preprint arXiv:2310.10012, 2023.
>
> [2] Yijun Yang. Mma-diffusion: Multimodal attack on diffusion models. In CVPR 2024.
>
> [3] Yimeng Zhang. To generate or not? safety-driven unlearned diffusion models are still easy to generate unsafe images... for now. In ECCV 2024.
>
> [4] Howard, Jeremy, and Sylvain Gugger. "Fastai: a layered API for deep learning." Information 11.2 (2020): 108.

---

### Official Review · Reviewer_RFw7 · 2025-11-03

**Soundness:** 3
**Presentation:** 3
**Contribution:** 3
**Rating:** 4
**Confidence:** 2

**Summary:**

The paper addresses a challenge of large-scale concept unlearning in text-to-image diffusion models. While existing approaches can remove individual concepts, they struggle when scaling to multiple concepts simultaneously, facing issues with conflicting weight updates, imprecise unlearning boundaries, and computational scalability bottlenecks.

**Strengths:**

1. The paper is well-structured and easy to follow
2. While the majority of other algorithms fine-tune the the entire model weights
3. Addresses fundamental challenges through theoretically grounded components
4. Comparison with other methods on a large-scale multi-concept unlearning (up to 50 concepts)
5. Extensive appendix with math proofs and detailed results

**Weaknesses:**

1. No comparison with parameter efficient Unlearning methods (e.g., SEMU https://arxiv.org/abs/2502.07587)
2. It would be beneficial to have the data from figure 3 in a from of a table in the appendix
3. Based on the theoretical foundation the method should also work on other SOTA models (e.g., SDXL, Stable Diffusion 3.5, FLUX.1-dev, Qwen-Image); however, no experimental confirmation is presented in the paper

**Questions:**

see Weaknesses section

---

> ### Author Response · Authors · 2025-11-19
>
> We sincerely thank the reviewer (RFw7) for the careful reading of our paper and for the constructive and insightful feedback. The comments greatly helped us improve the clarity, completeness, and technical rigor of the manuscript.
>
> **Weakness-1: Comparison with Parameter Efficient Unlearning Methods**
>
> The reason SEMU was not included in our original submission is that the work was only available as a preprint at the time of paper submission. We have now added SEMU to the Related Work (l128). In addition, we have incorporated SEMU into our experimental comparisons to ensure fairness and completeness. The tables below summarize the results of multi-concept unlearning (Imagenette[1] benchmark), explicit content unlearning (I2P benchmark) and artistic-style unlearning. Our method continues to outperform or remain highly competitive relative to SEMU and other baselines.
>
> Results for multi-concept unlearning:
> | Method| SD v1.5 | FMN  | SPM  | ESD  | MACE | SEMU | UCE  | RECE | SP   | **ScaPre** |
> |---|----|--|---|-|---|---|-|----|----|--|
> | **Avg Acc (↓)**    | 89.9    | 71.9 | 47.4 | 38.7 | 78.5 | 67.3 | 8.5  | 4.9  | 9.6  | **0.8** |
> | **CLIP_coco (↑)**  | 31.43   | 30.62| 30.81| 30.14| **31.02** | 30.58 | 29.45| 29.27| 29.25| 30.43 |
> | **UQ (↑)**         | ---     | 37.35| 49.89| 47.84| 35.07| 46.35| 37.23| 32.60| 31.78| **64.09** |
>
> Results for explicit content unlearning:
> | Method| FMN | AC  | UCE | ESD | SPM | MACE | SEMU | SP  | ScaPre | SD v1.4 |
> |---|-----|-----|-----|-----|-----|------|------|-----|----|---|
> | **Total ↓** | 448 | 838 | 182 | 315 | 229 | 111  | 207  | 89  | **54** | 743     |
> | **FID ↓** | 13.54 | 14.13 | 14.07 | 14.41 | 13.81 | **13.42** | 13.98 | 14.11 | 13.95 | 14.04 |
> | **CLIP ↑** | 30.43 | **31.37** | 30.85 | 30.69 | 31.24 | 29.41 | 30.14 | 30.79 | 30.88 | 31.34 |
>
> Results for artistic-style unlearning:
> | Method  | CLIP_art (↓) | CLIP_coco (↑) | CLIP_x (↑) | FID (↓) |
> |--|-|-----|---|-|
> | SD v1.5    | 31.25 | 31.43   | ---        | 13.60   |
> | FMN  | 30.67 | **31.20**  | 0.53       | 14.72   |
> | SPM| 29.40   | 29.73  | 0.33       | 22.75   |
> | ESD| 27.24 | 27.94  | 0.70 | 17.22   |
> | MACE   | 27.34 | 30.06    | 2.72   | **13.89** |
> | SEMU  | 28.64 | 29.87    | 1.23    | 23.92   |
> | UCE  | 26.95  | 28.21| 1.26   | 43.72   |
> | RECE | 25.94 | 26.86    | 0.92  | 49.32   |
> | SP  | 29.35  | 28.75   | -0.60 | 19.76   |
> | ScaPre | **26.51**    | 29.95          | **3.44**   | 14.37   |
>
> **Weakness-2: Table Representation of Figure 3 Data**
>
> We appreciate the suggestion and agree that including a table with the numerical values from Figure 3 enhances clarity. Accordingly, we'll add a detailed table with the exact data to the appendix.
> Moreover, we have added an analysis of how the execution time and peak memory usage change as more concepts are unlearned. The results show that while the execution time increases linearly with the number of concepts, the peak memory remains nearly constant with only minimal fluctuations.
> | Method  | FMN   | SPM    | ESD     | MACE  | UCE | RECE   | SP    | **ScaPre** |
> |----|---|---|--|---|---|---|---|-------|
> | **GPU Hours↓**         | 1.43  | 21.20  | 0.0917  | 1.94  | **0.017** | 0.0350 | 1.67  | 0.0317 |
> | **Memory (GB) ↓**     | 19.41 | 20.04  | 13.09   | 10.43 | 6.82 | 8.23   | 21.23 | **6.54** |
>
> | Number of concept  | 1  | 3| 10   | 20  | 30| 50|
> |---|----|-----|---|----|-----|-----|
> | **GPU Hours** | 0.0317  | 0.0951    | 0.317     | 0.634     | 0.951      | 1.585 |
> | **Peak Memory (GB)**   | 6.54     | 6.57      | 6.51  | 6.55  | 6.56   | 6.52  |
>
> **Weakness-3:  Extension to SDXL and Flux 1.0**
>
> We fully agree that the theoretical foundation suggests that the proposed approach should extend beyond Stable Diffusion v1.5. We conduct additional experiments (multi-concept unlearning and artistic-style unlearning) on SDXL and Flux1.0, two widely adopted modern architectures, to empirically validate the effectiveness of our method on stronger generative backbones. Furthermore, we also include comparisons against the UCE baseline, which also supports SDXL and Flux1.0, to provide a fair and rigorous evaluation.
> | Imagenette classes | SDXL | SDXL–UCE | SDXL–ScaPre | Flux1.0 | Flux1.0–UCE | Flux1.0–ScaPre |
> |--|---|--|-|-|--|---|
> | **Avg Acc (↓)**    | 97.9 | 0  | 7.2 | 96.2  | 0 | 9.4  |
> | **CLIP_coco (↑)**  | 32.74| 22.8  | 30.17  | 32.65 | 21.9   | 38.7|
> | **UQ (↑)**  | --   | --   | 43.5 | --  | --   |46.7 |
>
> | Method | SDXL | UCE  | ScaPre| Flux1.0 | Flux1.0–UCE | Flux1.0–ScaPre |
> |---|---|--|---|---|----|-|
> | **CLIP_art (↓)**     | 32.47 | 25.38 | 27.36  | 32.18   | 26.23  | 27.67 |
> | **CLIP_coco (↑)**    | 31.98 | 24.98 | 30.96  | 32.43  | 24.84  | 31.04 |
> | **CLIP_x (↑)**       | --    | -0.4  | 3.6   | --      | -1.39 | 3.37   |
> | **FID (↓)**          | 9.12  | 52.6  | 11.4   | 10.31   | 49.71  | 13.65  |
>
> [1] Howard, Jeremy, and Sylvain Gugger. "Fastai: a layered API for deep learning." Information 11.2 (2020): 108.

---

### Author Response · Authors · 2025-11-22

We sincerely thank all Area Chairs and reviewers for their efforts in handling our manuscript. We have carefully considered all the valuable comments and have addressed every issue raised in the reviews within the rebuttal, incorporating corresponding revisions to resolve the major concerns. A further refined and fully polished revised version of the manuscript will be provided shortly. We once again express our sincere appreciation for the Area Chairs’ and reviewers’ time and commitment to the review process.

---

### Author Response · Authors · 2025-11-26

Dear ACs,

Once again, we would like to express our sincere thanks to you and to reviewers for the time and effort invested in evaluating our submission. We truly appreciate the constructive feedback provided.

We would like to let you know that we have carefully addressed all reviewer comments and have revised the paper accordingly. Our rebuttal and the related clarifications have been fully prepared with the intention of being as helpful and transparent as possible.

Since we have not yet received any follow-up from the reviewer(s), we were wondering whether it might be possible for you to kindly remind them at your convenience. We would greatly appreciate any additional feedback they may be willing to provide, as it would help ensure that our responses are properly understood and considered during the final evaluation.

Thanks again for all ACs' and reviewer' time and commitment to the review process.

---

### Author Response · Authors · 2025-12-01

Dear ACs,

We sincerely thank you for handling our submission.

After carefully analyzing all reviewer comments, we find that the concerns tend to cluster around three core issues. We have fully addressed each of these issues in our rebuttal.

First, reviewers emphasized the need to validate the generalization capability of our approach beyond SD v1.4 and SD v1.5 and to demonstrate its applicability to stronger and more modern diffusion architectures. In response, we conducted new experiments on SDXL and FLUX.1-dev. These results confirm that our method generalizes effectively to these stronger models without the need for additional adjustments.

Second, reviewers requested additional clarity regarding Mutual Information (MI) related implementation details. This includes how target and neutral inputs are constructed, how thresholds and sample sizes are selected, how stable the MI estimates are across layers, and which cross attention layers and which key and value branches are edited. We have provided detailed explanations, expanded illustrations, and additional ablations to make all aspects of the design fully transparent.

Third, reviewers pointed out missing information on dataset construction. This includes data splits and the way prompts are constructed. We have provided full responses to all of these points and clarified both aspects in detail.

These three main concerns, together with all remaining issues raised by the reviewers, have been fully resolved through our clarifications and additional experiments. We sincerely appreciate the efforts of both the ACs and the reviewers, and we hope that our responses satisfactorily address every concern that was raised.

---

### Meta-Review · Area_Chair_GR2e · 2026-01-06

**Summary:**

Reviewers appreciated that the proposed approach combines a conflict-aware stable design and an Informax Decoupler, effectively addressing problems of instability in large-scale unlearning and imprecise separation of target/non-target concepts. Reviewers also highlighted  empirical validation,  light-weight design, benchmark construction, and convex quadratic core with unique Sylvester solution as some of the strong points. Reviewers also raised some concerns in the initial reviews including, comparisons with stronger and more modern diffusion architectures, missing implementation details (e.g., clarity regarding Mutual Information (MI)), and missing details on dataset construction.

**Reviewer Concerns:**

The meta-reviewer believes that the authors provided a comprehensive rebuttal. For instance, the authors provided a comparison with  parameter efficient unlearning methods (including the recent SEMU). The comparisons are provided for multi-concept unlearning, explicit content unlearning and artistic-style unlearning. In the rebuttal, authors also provided results (multi-concept unlearning and artistic-style unlearning) with SDXL and Flux1.0, dataset construction details, and an analysis on scaling trends and computational effiency.

**Reviewer Scores:**

Since the main concerns such as, comparison with  parameter efficient unlearning methods, additional comparisons with SDXL and FLUX1.0, and missing dataset construction details are comprehensively addressed in the rebuttal, the meta-reviewer believes the reviewers will have a better positive outlook if they had been able to participate fully in the discussion.

---

### Decision · Program_Chairs · 2026-01-26

Accept (Poster)